# Energy Model for Long-Term Scenarios in Power Sector under Energy Transition Laws

**Gabriela Hernández-Luna** [1], **Rosenberg J. Romero** [1], **Antonio Rodríguez-Martínez** [1,*], **José María Ponce-Ortega** [2], **Jesús Cerezo Román** [1] and **Guadalupe Diocelina Toledo Vázquez** [1]

1   Engineering and Applied Sciences Center Research—CIICAp, Autonomous University of Morelos State—UAEM, Av. Universidad 1001, Chamilpa, Cuernavaca 62210, Morelos, Mexico; gabriela.hernandez@uaem.mx (G.H.-L.); rosenberg@uaem.mx (R.J.R.); jesus.cerezo@uaem.mx (J.C.R.); diocelina.toledo@gmail.com (G.D.T.V.)

2   Faculty of Chemical Engineering, Universidad Michoacana de San Nicolás de Hidalgo, Av. Francisco J. Múgica S/N Ciudad Universitaria, Morelia 58030, Michoacán, Mexico; jmponce@umich.mx

*   Correspondence: antonio_rodriguez@uaem.mx; Tel.: +52-777-3297-084 (ext. 6262)

**Abstract:** High electricity demand, as well as emissions generated from this activity impact directly to global warming. Mexico is paying attention to this world difficulty and it is convinced that sustainable economic growth is possible. For this reason, it has made actions to face this problem like as launching constitutional reforms in the power sector. This paper presents an energy model to optimize the grid of power plants in the Mexican electricity sector (MES). The energy model considers indicators and parameters from Mexican Energy Reforms. Electricity demand is defined as a function of two population models and three electricity consumption per capita. Prospectives are presented as a function of total annual cost of electricity generation, an optimal number of power plants—fossil and clean—as well as $CO_2$eq emissions. By mean of the energy model, optimized grid scenarios are identified to meet the governmental goals (energy and environment) to 2050. In addition, this model could be used as a base to identify optimal scenarios which contribute to sustainable economic growth, as well as evaluate the social and environmental impacts of employed technologies.

**Keywords:** electricity model; power plants prospectives; Mexican prospectives

## 1. Introduction

Power in any region of the world is essential for development and economic growth. The electricity sector has historically developed under uncertainty and constant changes, especially since the end of the 1980s when the damage to the ozone layer was evidenced [1]. The magnitude of the event gave a global social concern, which is reflected in the Montreal Protocol in 1987 [2], the multilateral treaty on the environment has had the most success in all of history. This treaty subsequently gives rise to international agreements such as the Kyoto Protocol in 1998 [3] until reaching the Sustainable Development Goals of the United Nations launched in 2015 [4], also considering the Paris Agreement of 2015 [5] which established keeping the global temperature rise below 2 °C. However, despite the international effort, electric consumption continues, increasing incessantly, as reported by the World Bank in 2017 [6], as can be seen in Figure 1.

This high electricity demand brings an environmental impact, which has reached 30% of the total global emissions of greenhouse gases (GHG) as Figure 2 shows.

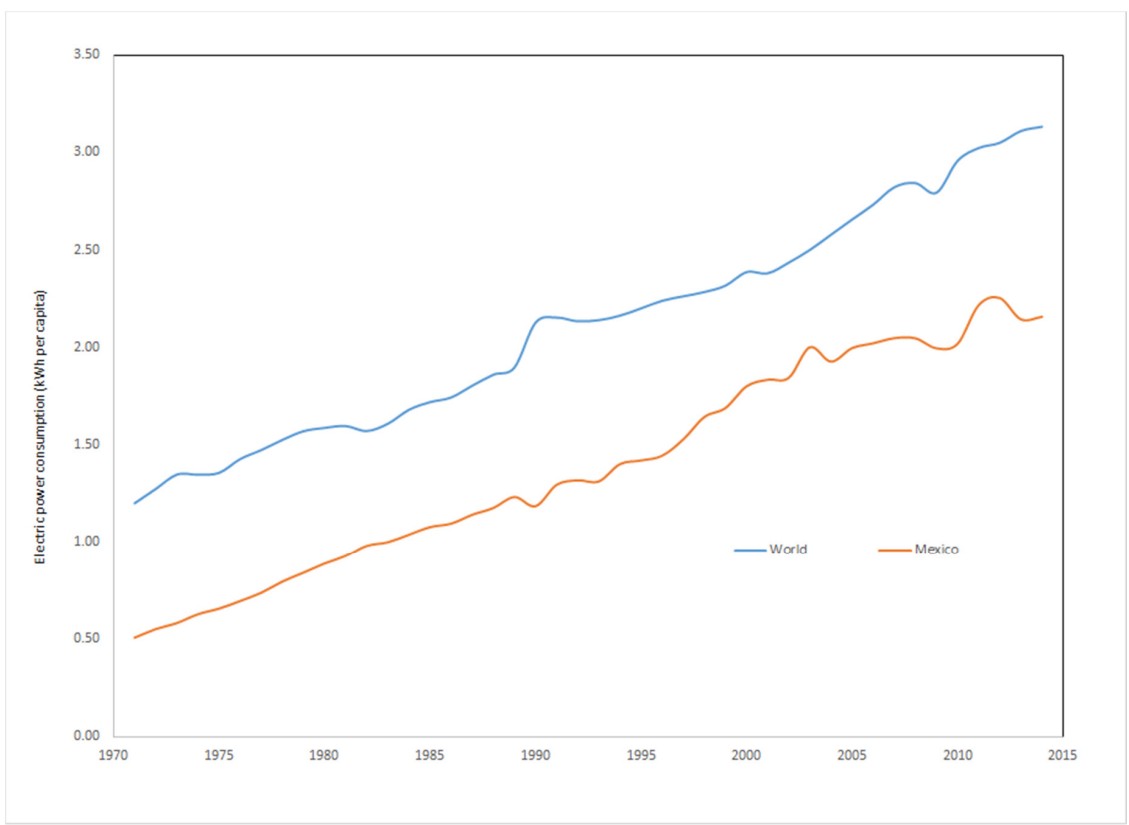

**Figure 1.** World and Mexico electric power consumptions. Prepared by the authors based on data from cited Reference [6].

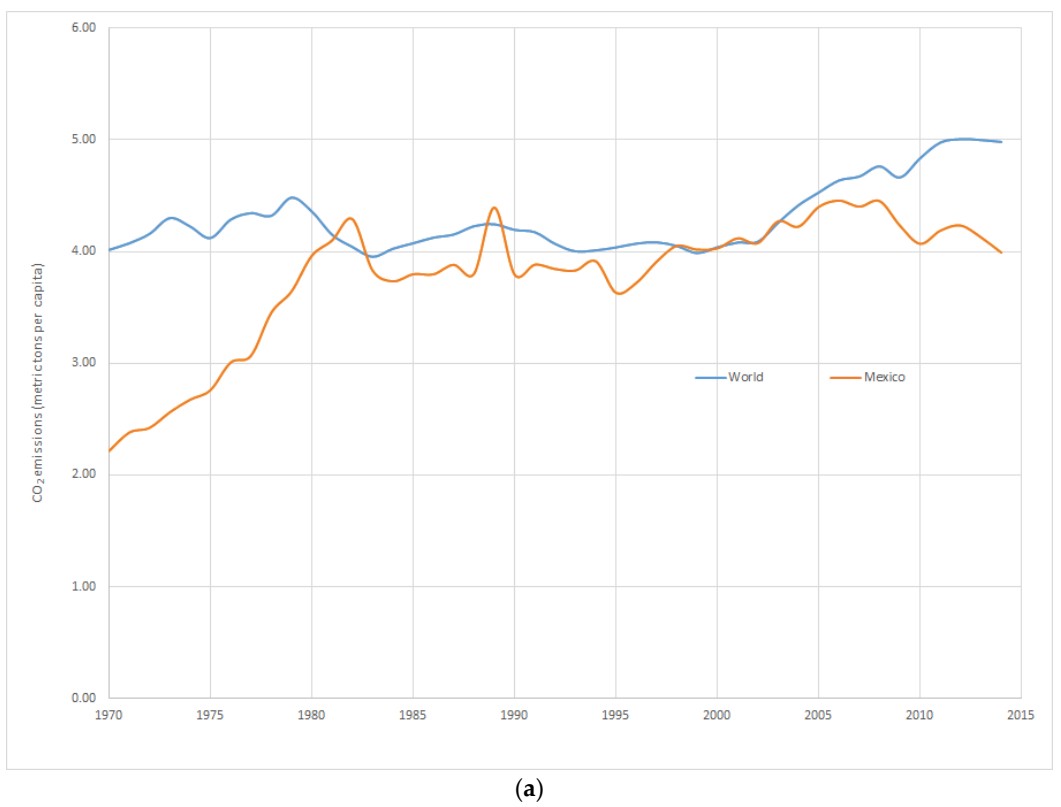

(**a**)

**Figure 2.** *Cont.*

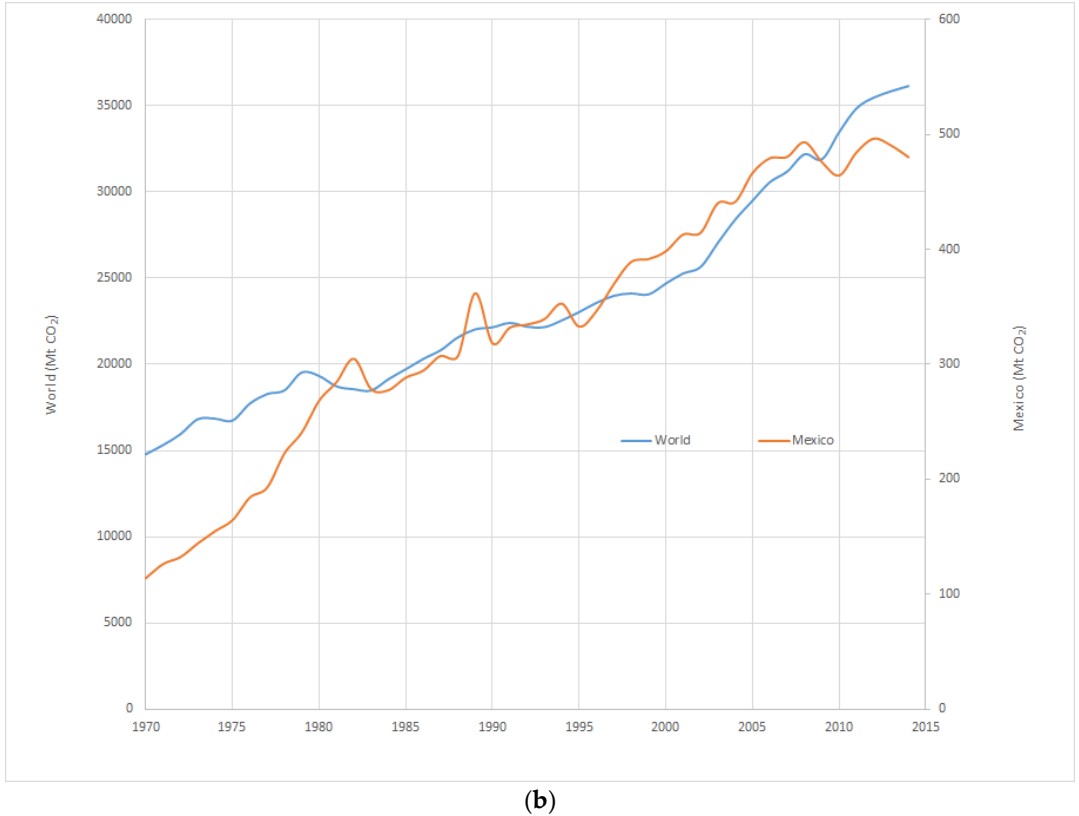

(**b**)

**Figure 2.** World and Mexico (**a**) $CO_2$ emissions per capita and (**b**) $CO_2$ total emissions. Prepared by the authors based on data from cited Reference [6].

The world faces a challenge between meeting the world's electricity demand and reducing greenhouse gas emissions produced by this sector. Mexico is not only sensitive to this global reality but it is also convinced that economic progress can and should be parallel to environmental protection. Under this premise, the Mexican government has taken actions to face this challenge as issuing reforms related to the energy sector, specifically in the Mexican Electricity System (MES) since this is the second largest emitter of GHG in the country, only after the transport sector. The GHG emissions produced by electricity sector has reached approximately 25% of total national emissions in 2015, as reported by National Institute of Climate Change (INECC) in 2015 [7] (see Figure 3) due to electricity generated in Mexico which is around 80% from fossil energy sources as Energy State Secretary (SENER) indicated in 2017 [8].

The amendments or reforms in energy affairs are also a product of the influence of international organizations and financial institutions as well as of changes in the price of fuels [9]. The main reforms launched in this field are "*General Climate Change Law*" (GCCL) in 2012 [10] and the "*National Climate Change Strategy*" (NCCS) in 2013 [11], whose main objective is minimizing GHG emissions produced by electricity generation and also modernize the Mexican Electricity System. The strategy considers the planning and sustainable growth, low carbon emissions of the MES in long term, reflected in an increase of the so-called "clean technologies" (According to the Energy Transition Law, 2015 [12]) for electricity generation. The goal is to achieve at least 35% of electricity generated from clean sources for the year 2024, in addition in achieving a specific GHG emission reduction of 30% and 50% for the years 2020 and 2050, respectively, with respect to the 2000 year emissions [10,11].

In response to this questioning, the use of renewable technologies is promoted to replace the main conventional technologies and achieve the proposed goals, however, this option can lead to risks such as potential incidences in the planning and construction of plants generating power, in addition to potential situations that would increase the start-up time of the electricity generating plants. This is

one of the reasons why the use of simulations and numerical models can be a useful tool that allows strategically planning and visualizing the sector to suggest scenarios where proposed objectives by reforms are achieved.

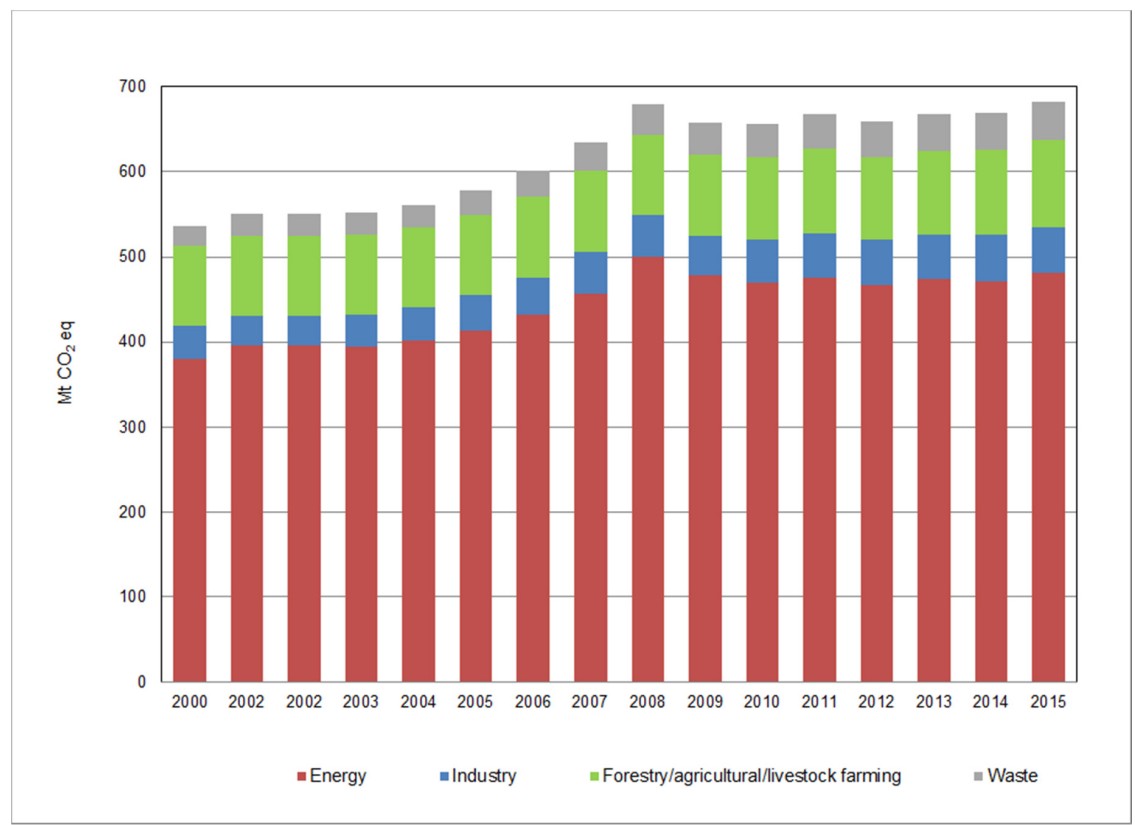

**Figure 3.** Mexican $CO_2$ emissions by sector. Prepared by the authors based on data from cited Reference [7].

The energy sector has been the object of study from various points of view; from the economic, technical as well as environmental. This section presents a brief review of the research.

Manzini et al. [13] analyzed the environmental impacts of the use of renewable sources in electricity generation. They basically identified $CO_2$, $CH_4$, NOx and SOx emissions for conditions where renewable energy reaches a contribution of 31 and 43% of installed capacity.

The World Bank [14] has also studied this sector and in 2009 they conducted an analysis to identify areas of influence in the reduction of GHG; to identify the most important to the electricity sector. In this study, the authors also identify the minimum costs available to be implemented using the cost-benefit methodology.

This cost-benefit methodology has been used by Lund and Mathiesen [15] for the study of the electrical system in Denmark, analyzing transition scenarios emphasizing areas such as energy efficiency, reduction of emissions of $CO_2$eq and industrial development, whose objective was is to reach up to 100% of the demand for electrical energy using renewable sources.

In 2014, Santoyo-Castelazo et al. [16] analyzed the environmental implications of decarbonization of the MES, by proposing scenarios with diverse configurations of the energy matrix of technologies used for combined-cycle power plant with life cycle analysis studies. The authors concluded that it is evident the possibility of reducing the environmental impact with the proposals made.

The MES energy transition has been developing also studied by Vidal-Amaro et al. [17]. The authors made a proposal to determine an optimal configuration of the energy matrix consisting of fossil and renewable sources for one moment in time, 2024 with the goal of reaching 35% of electricity

from renewable sources, besides the authors managed to identify an optimal configuration as well as quasi-optimal configurations as an alternative.

The review focuses its efforts on analyzing GHG emissions as well as on identifying or evaluating the contributions of renewable energy sources to the MES in a specific scenario; however, no methodologies have been identified that propose identifying the number of power plants generating electricity to meet the demand neither short nor long term. This paper presents a proposal for an alternative electricity generation model to optimize the number of power plants to satisfy demand under Mexico's government policies with at short and at long term environmental impact scenarios.

## 2. Methods

This paper proposes an energy model to satisfy the short and long term demand for power plants in periods of lustrum from 2020 to 2050 year at several scenarios and it is compared with International Energy Agency (IEA) [18] data. The objective function is to minimize the Total Annual Cost (TAC) of electricity generation as well as determine a matrix made up of a number of electricity generating power plants required, minimizing power plants number whose primary energy source is from fossil resources. The proposed model considers population growth as a function to determine the electricity demand, taken from the World Population Prospects [19] as can be seen in Figure 4.

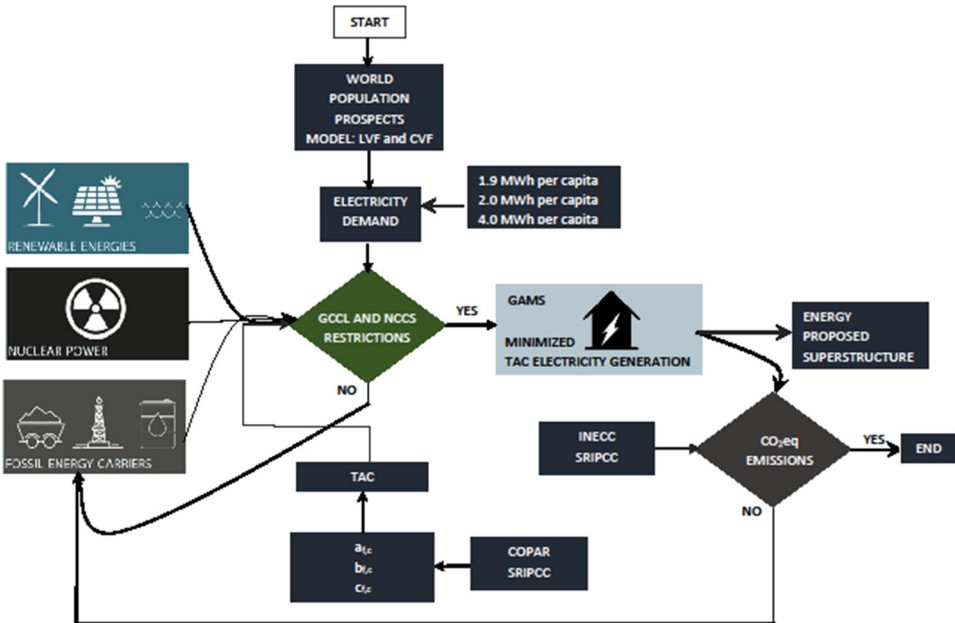

**Figure 4.** Schematic representation of Mexican Electricity System (MES) optimization.

The proposed matrix (composed by the number of power plants) is based on a superstructure which considers technologies based on fossil and clean energy sources for electricity generation, presented in Figure 4 and the code is in Appendix A. In addition, the software GAMS © identifies the optimal number of power plants generation to satisfy a period demand and $CO_2$eq emissions. The $CO_2$eq emissions are compared with NCCS and GCCL, if the value is lower than the constraint then the superstructure is shown as a result.

Electricity demand is determined using two population models from the United Nations (2017) [19], which are detailed next.

## 2.1. Electricity Demand

The proposed mathematical model assumes that the annual electricity demand is satisfied with electricity generation from two main energy sources: from fossil and clean sources, which is stated as follows:

$$E_d = \sum_i E_i \tag{1}$$

$E_d$ is the annual electricity demand to be met, $E_i$ the total electricity generated for each analyzed period corresponding to the total amount of the electricity coming from fossil sources, $E_f$, as well as from clean sources, $E_c$, in other words, $i = f$ for fossil sources, while $i = c$ for clean sources.

Electricity demand, $E_d$, is defined with two population models used in the realization of World Population Prospects of the Department of Economics and Social Affairs of the United Nations [19]. The population models are the Low Variability of Fertility (LVF) and life expectancy at birth and Constant Variability of Fertility (CVF) and life expectancy at birth. Three demand conditions per capita are assumed in each model: 1.9, 2.0 and 4.0 MWh. The demand of 1.9 MWh per capita is taken from the historical demand of Mexico in the period of 2000 to 2010; the demand of 2.0 MWh per capita considers an increase of 10% in the population demand and finally, the demand of 4.0 MWh per capita is the typical demand of the population consumption from developed countries. Table 1 presents a matrix with the described conditions.

**Table 1.** Matrix of described conditions.

| | | | | | | |
|---|---|---|---|---|---|---|
| **Population Model** | | | | | | |
| | **LVF** | | | **CVF** | | |
| **Electricity Demand [MWh per capita]** | | | | | | |
| Year | 1.9 | 2.0 | 4.0 | 1.9 | 2.0 | 4.0 |
| 2015 | LVF-1.9-2010 | LVF-2.0-2010 | LVF-4.0-2010 | CVF-1.9-2010 | LVF-2.0-2010 | LVF-4.0-2010 |
| 2020 | LVF-1.9-2010 | LVF-2.0-2010 | LVF-4.0-2010 | CVF-1.9-2010 | LVF-2.0-2010 | LVF-4.0-2010 |
| 2025 | LVF-1.9-2025 | LVF-2.0-2025 | LVF-4.0-2025 | CVF-1.9-2025 | LVF-2.0-2025 | LVF-4.0-2025 |
| 2030 | LVF-1.9-2030 | LVF-2.0-2030 | LVF-4.0-2030 | CVF-1.9-2030 | LVF-2.0-2030 | LVF-4.0-2030 |
| 2035 | LVF-1.9-2035 | LVF-2.0-2035 | LVF-4.0-2035 | CVF-1.9-2035 | LVF-2.0-2035 | LVF-4.0-2035 |
| 2040 | LVF-1.9-2040 | LVF-2.0-2040 | LVF-4.0-2040 | CVF-1.9-2040 | LVF-2.0-2040 | LVF-4.0-2040 |
| 2045 | LVF-1.9-2045 | LVF-2.0-2045 | LVF-4.0-2045 | CVF-1.9-2045 | LVF-2.0-2045 | LVF-4.0-2045 |
| 2050 | LVF-1.9-2050 | LVF-2.0-2050 | LVF-4.0-2050 | CVF-1.9-2050 | LVF-2.0-2050 | LVF-4.0-2050 |

## 2.2. Model Formulation

Electricity generation from each type of fuel is determined with Equation (2).

$$\sum E_i = \sum_f \sum_c E_{f,c} \leq \sum_f \sum_c \left( E_{f,c_{inst}} + x_{f,c} \cdot E_{f,c_{cap}} \cdot fc \right) \tag{2}$$

The fossil energy sources, $i = f$, consider electricity generation from coal, *co*; diesel, *d*; gas *g* and fuel oil *o*, while electricity generation from clean energy sources, $i = c$, considers the use of biomass, *bm*; carbon capture and storage, *ccs*; eolic, *eo*; photovoltaic, *phv*; geothermal, *gtr*; hydraulic, *hdr*; nuclear, *nc* and solar concentration, *sc*.

The proposed model determines superstructure electricity generation in each analyzed period taking into account electricity contribution generated in a period prior to the analyzed one, $E_{f,c_{ins}}$, the number of power plants generation $x_{f,c}$ and the capacity of each power plant, $E_{f,c_{cap}}$ as well as a coefficient fc to homogenize units.

Total country annual cost of electricity generation from each energy plant as well as the emissions due to this process is determined by Equations (3) and (4), respectively.

$$TAC(E_i) = \sum_f \sum_c \left( E_{f,c} \cdot Cost \left( a_{f,c} + b_{f,c} + c_{f,c} \right) \right) \tag{3}$$

where $a_{f,c}$ represents investment costs, $b_{f,c}$ is the associated cost from fuel type used and finally $c_{f,c}$ represents operation and maintenance cost, considering fixed and variable costs, taken from Generation, Costs and Reference Parameters for the Formulation of Investment Projects for the Mexican Electricity Sector (COPAR) [20] and Special Report of the Intergovernmental Panel of Climate Change (SRIPCC) of 2012 [21].

$CO_2eq$ emissions are determined with the Equation (4).

$$CO_2eqEm(E_i) = \sum_f \sum_c \left( E_{f,c} \cdot CO_2eq_{f,c} \right) \tag{4}$$

$COeq_{f,c}$ factor is particular for each type of fuel, taken from the INECC [7] and SRIPCC [21].

In an integrated manner, for each analyzed electricity demand condition, TAC of optimized electricity generation, matrix formed by the number of power plants required to satisfy the demand, integrated by power plants that use fossil fuel technologies and clean, as well as the emissions generated by each type of technology are calculated.

## 2.3. Solution Strategy

To determine the set of optimal solutions that satisfy governmental criteria in each electrical demand condition, epsilon constraint method [22] is implemented in the proposed model. The Pareto chart is constructed using that demand condition and optimized TAC. The epsilon constraint method satisfies governmental objectives proposed in the GCCL and NCCS, (2012) and (2013), respectively. These restrictions have implications to be considered in the application of the model, presented in Equations (5) and (6).

$$E_f < \%f \cdot E_d \tag{5}$$

$$E_c < \%c \cdot E_d \tag{6}$$

$E_f$ is the amount of electric energy generated from fossil sources, $E_c$ is the amount of electric energy generated from clean sources, described in Table 2 and in Figure 5.

**Table 2.** Objectives and goals to electricity generation by source energy defined in the General Climate Change Law (GCCL) Prepared by the authors based on public data from cited Reference [10].

| Electricity Generation Percentage by Energy Source, % | | |
|---|---|---|
| Year | Fossil, $f$ | Clean, $c$ |
| 2024 | 65 | 35 |
| 2035 | 60 | 40 |
| 2050 | 50 | 50 |

The model considers linear tendency and it was formulated in the General Algebraic Modelling System (GAMS) software used for modelling and mathematical optimization. The results obtained for each condition is shown in Table 1 and under the restrictions presented in Table 2 are detailed in the next section.

The restrictions are used in MES planning scenarios throughout the analysis period.

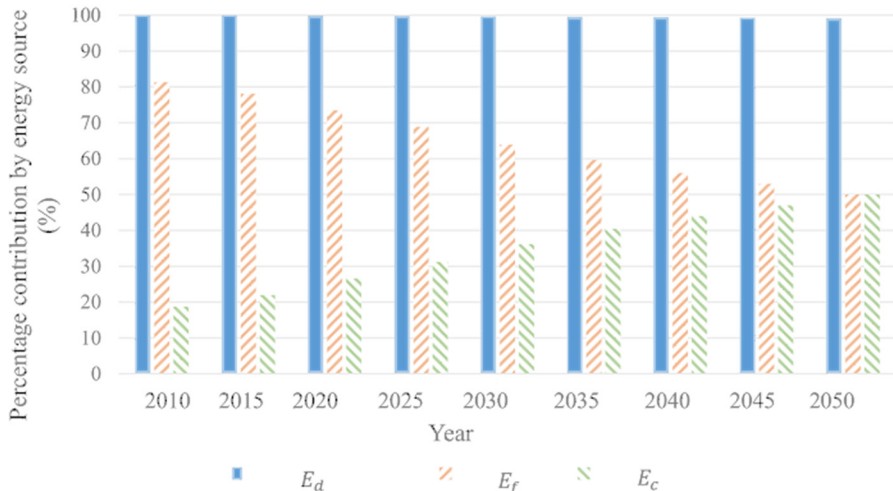

**Figure 5.** Governmental objectives to satisfy in General Climate Change Law (GCCL). Prepared by the authors based on data from cited reference [10].

## 3. Results

The modelling scenarios are presented in each electricity demand condition at three different conditions: electricity generation costs, optimized power plants number and generated emissions by this activity.

### 3.1. Electricity Demand

Electricity demand conditions determined with two population models throughout the analyzed period is presented in Figure 6. Three analyzed demand conditions using LVF presents a growth rate of 13% during all period, meanwhile the corresponding three analyzed demand conditions growth rate using CVF model is 50% are shown.

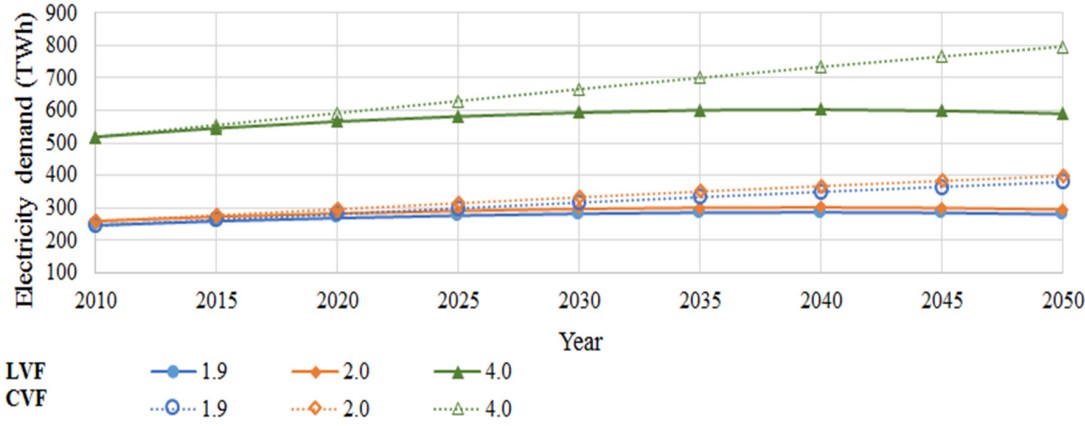

**Figure 6.** The analyzed electricity demand conditions determined with two population models and three demand conditions per capita. Prepared by the authors based on data from cited reference [19].

Electricity demand for both population models shows similar behavior, at first sight, both vary directly with time, however, it is possible to identify differences. Electricity demand determined with the LVF model presents three different stages throughout the period of analysis. From the beginning of the analysis until the year 2030 a constant growth is observed, followed by a stage without growth until the year 2045; finally, in the last stage of the period, the modelling shows a demand deceleration. The electric demand with the CVF model shows a constant behavior throughout the analysis period, basically an incessant increase.

### 3.2. TAC of Electricity Power Generation and Optimized Centrals Number

Once the energy demand to be supplied has been defined throughout the analyzed period, the modelling takes the TAC of electric power generation and number of plants for takes two population models (LVF and CVF) and three demand conditions (1.9, 2.0 and 4.0 MWh per capita). In order to determine the number of power plants, priority is given to those that use clean technologies and minimize those of fossil technologies.

Optimized TACs of electricity power generation compute differences between the two population models. The corresponding one from demand determined with LVF population model shows an almost constant growth, however, the demand decreases in the last period of the analysis, specifically in the last two decades. Maximum electricity demand is located in years 2035 and 2045, after this period, electricity demand has a decreasing tendency. This behavior is observed for the three demand conditions.

Optimized TACs present a different behavior using the CVF model—it grows directly proportional to electricity demand during the entire analyzed period. Details for the scenarios using both population models are presented next.

#### 3.2.1. LVF Population Model

Details for the electricity demand determined using LVF population model as a function of TAC and optimized power plants number are analyzed and classified as fossil fuel and clean fuels of energy supply in Figure 7. Total electricity demand is presented by (T), electricity demand from fossil fuels are represented by (F) and electricity demand from clean fuels are represented by (C). Total electricity demand exhibit positive growth behavior meanwhile electricity demand from fossil technologies show an opposite trend as a function of time. Clean technologies have a constant growth meaning the main energy technologies to be employed are from clean energy sources.

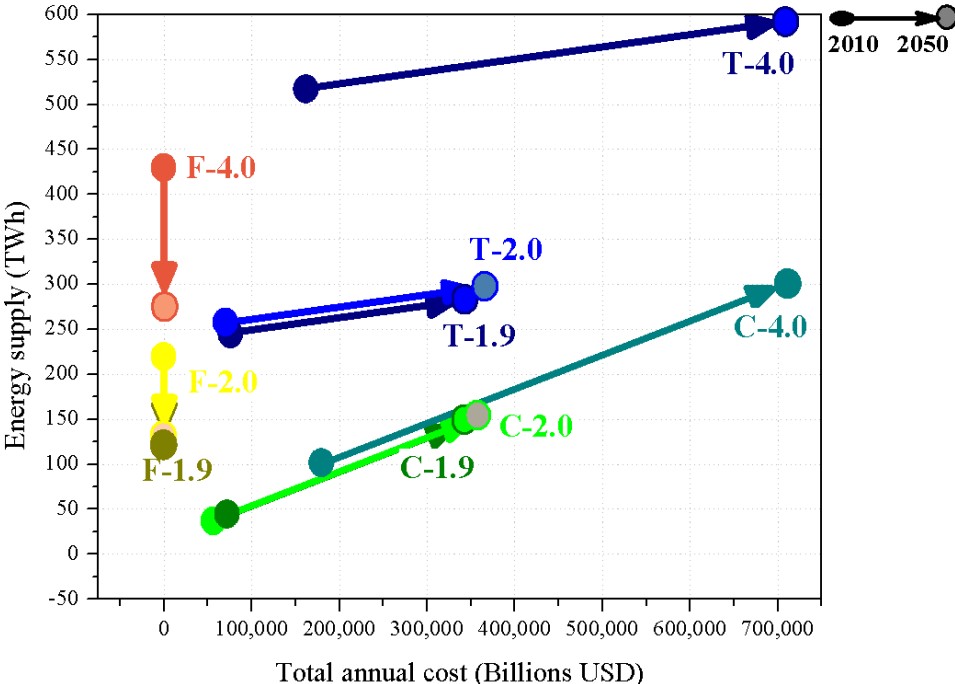

**Figure 7.** Scheme of the optimized power plants generation centrals number minimizing Total Annual Cost (TAC) of electricity generation for Low Variability of Fertility (LVF) population model [19] and three electricity demand conditions per capita. Prepared by the authors based on data from cited reference.

Clean energy electricity demand shows a growth rate of 204.37%, meanwhile electricity demand from fossil energy sources has a decrease of 30.04% at the end of analyzed period respect year 2010.

If fossil fuel energy (F) does not have new investment, just maintenance, then the value in Figure 7 is relatively low, compared with that of clean energy, so F looks like 0, until the value is 2.1E + 08.

The case of the electricity demand of 1.9 KWh per capita shows total electricity supply from 2010 to 2050 initializing at 2.46 TWh until it rises to 2.80 TWh at the end of the analyzed period. This energy requirement impacts directly in the number of electric power plants necessary to satisfy this demand, presenting a growth rate from 11.27% and 99.51% from fossil and clean energy sources, respectively, at the end of the analyzed period.

Electricity demand of 2.0 KWh per capita, shows that the electricity supply in the analyzed period starts at 2.60 TWh until it raises 2.95 TWh. This energy requirement impacts directly in the number of electric power plants necessary to satisfy this demand, which presents a growth rate from 11.06% and 75.69% from fossil and clean energy sources.

The Electricity demand of 4.0 KWh per capita, shows electricity to supply from from 2010 to 2050 initializing with 5.18 TWh until it raises 5.90 TWh. It is important to highlight that even though this is a necessity, electricity demand presents a growth rate of 9.38% and 74.98%, from fossil and clean energy sources, respectively.

Optimized matrix of number of electricity generating power plants from fossil and clean energy sources to satisfy the demand of 1.9 MWh per capita are presented in Figures 8 and 9.

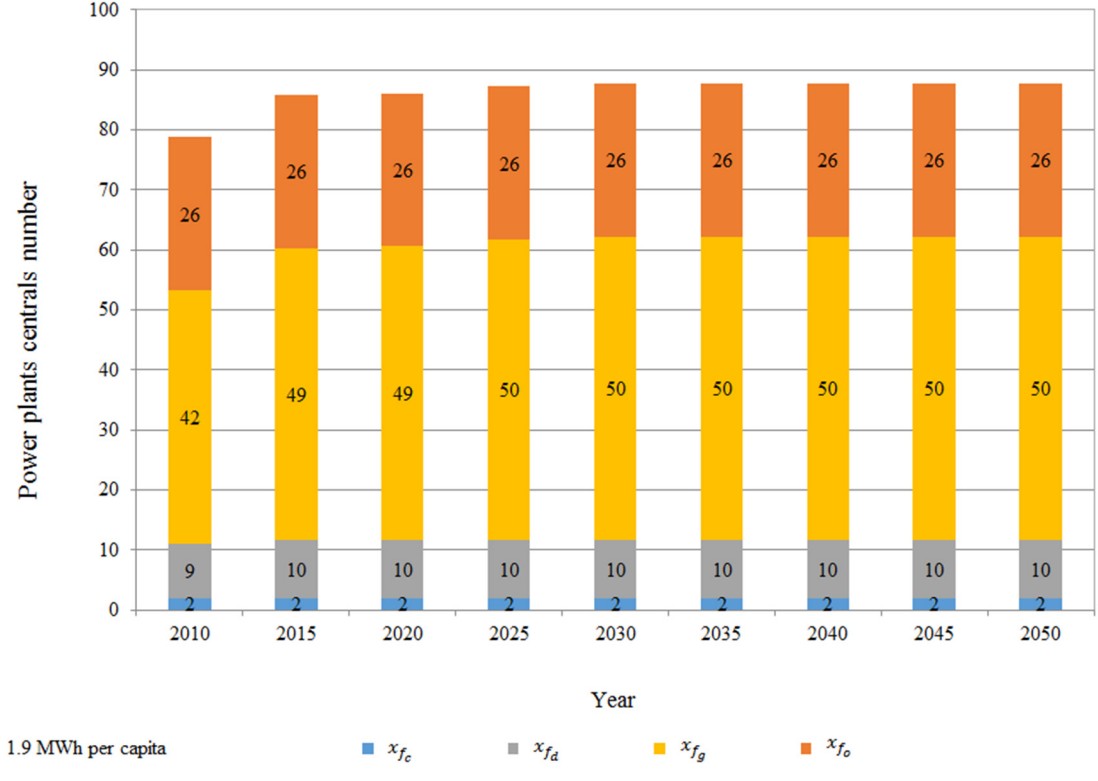

**Figure 8.** Optimized power plants number from fossil energy resource for 1.9 MWh per capita demand with the LVF model.

As can be appreciated from Figure 8, optimized fossil power plants' number for demand electricity condition of 1.9 MWh per capita, presents a quasi-constant behavior during the analyzed period; only a minimum growth is observed from gas power plants number.

The optimized power plants' number from clean sources presents discrete but constant growth during the whole analyzed period. Power plants which use technology from bioenergy source present higher growth, although their contribution is modest. Hydraulics is a technology source with the greatest contribution within the energy matrix.

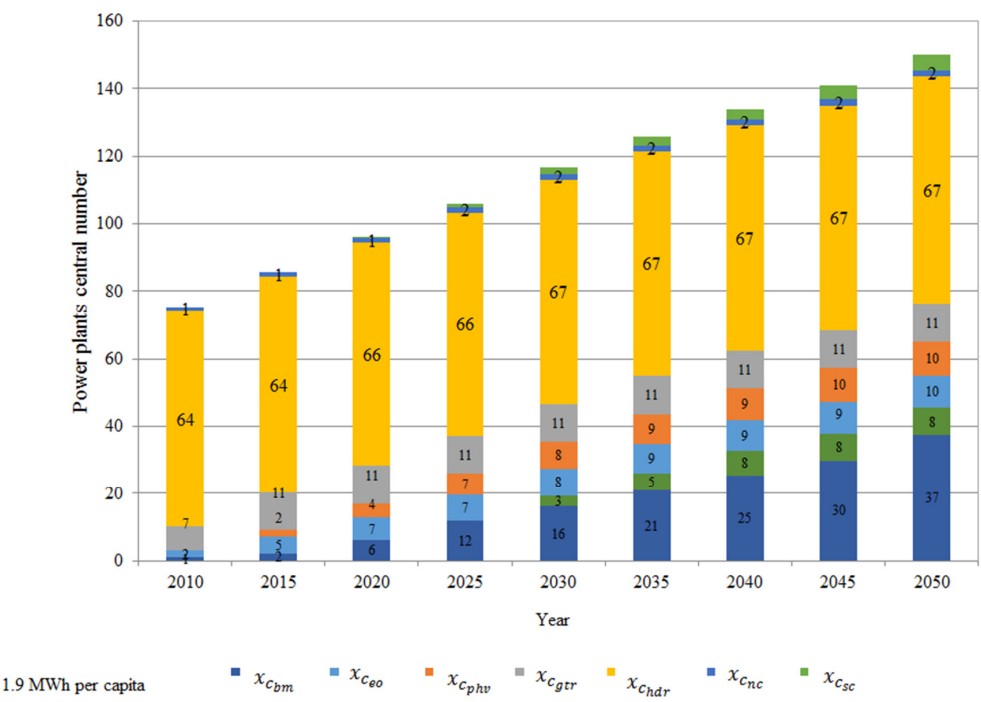

**Figure 9.** Optimized power plants number from clean energy resource for 1.9 MWh per capita demand with LVF model.

The number of electricity demand per capita of 2.0 MWh of optimized fossil and clean power plants is presented in Figures 10 and 11. The model shows that the optimized fossil power plant's number remains almost constant during the whole analyzed period, meanwhile the optimized clean power plant's number present a discrete growth.

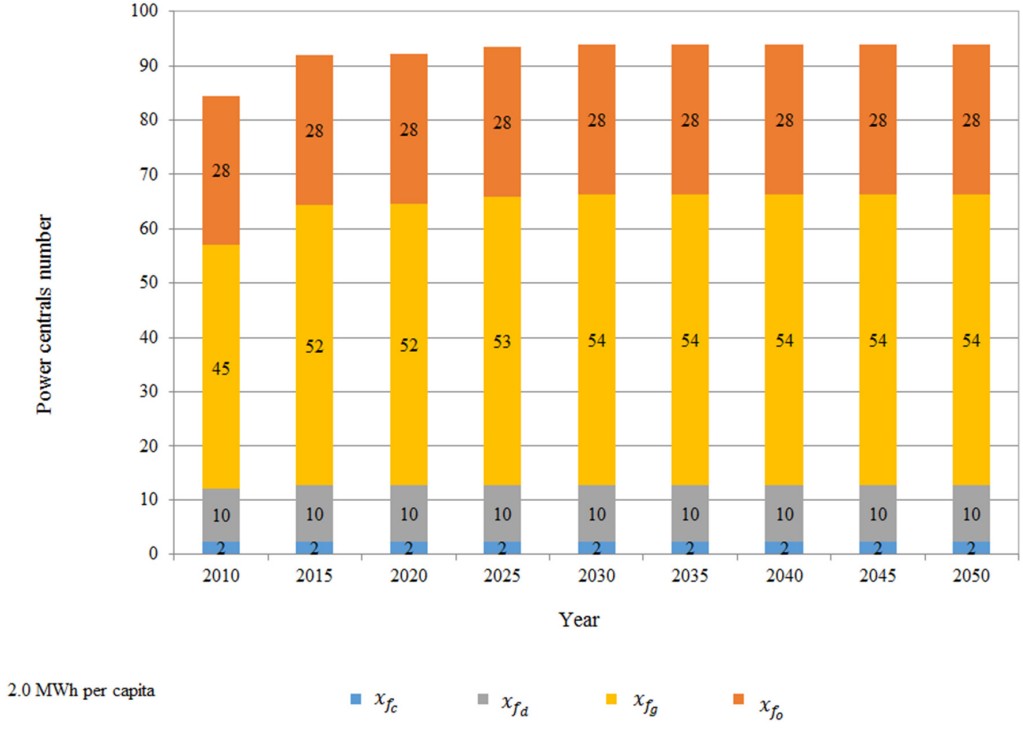

**Figure 10.** Optimized power plants number from fossil energy resource for 2.0 MWh per capita demand with the LVF model.

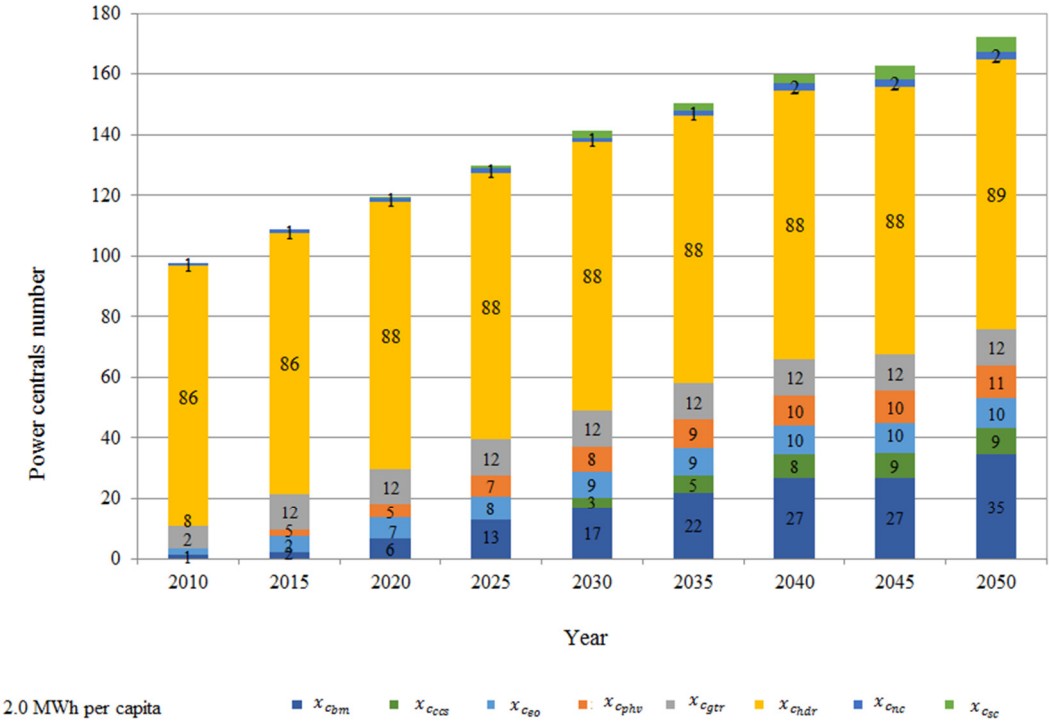

**Figure 11.** Optimized power plants number from clean energy resource for 2.0 MWh per capita demand with the LVF model.

Optimized fossil and clean power plants number of electricity demand per capita of 4.0 MWh are presented in Figures 12 and 13.

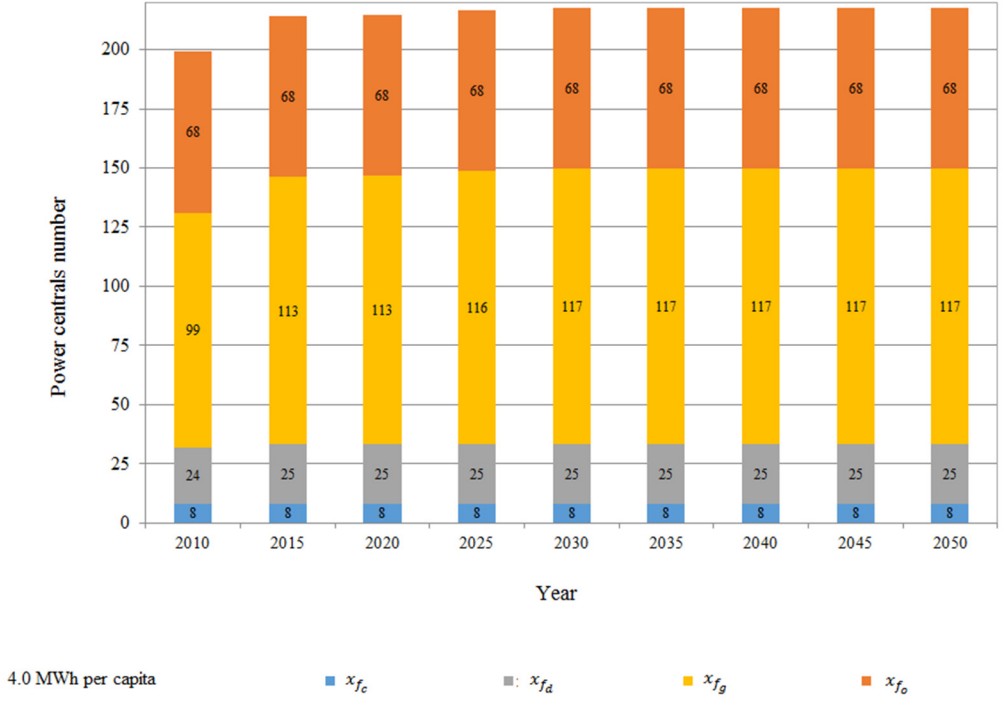

**Figure 12.** Optimized power plants number from fossil energy resource for 4.0 MWh per capita demand with the LVF model.

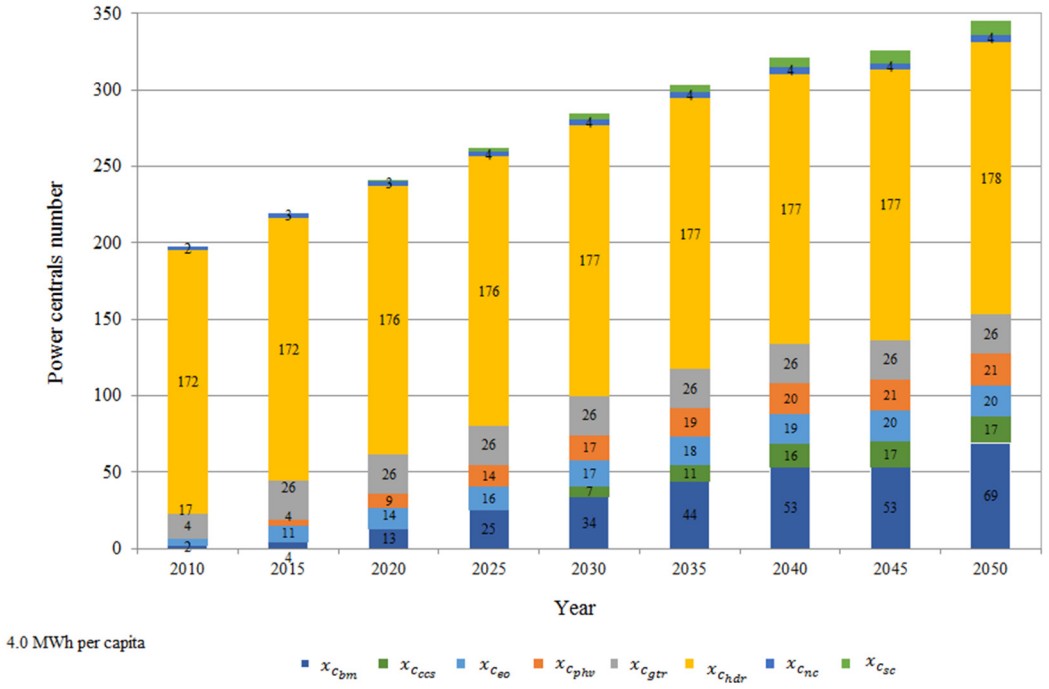

**Figure 13.** Optimized power plants number from clean energy resource for 4.0 MWh per capita demand with the LVF model.

In this electricity demand case, fossil power plants present an almost constant behavior, contrary to clean energy sources for which growth is constant, discrete but constant. Power plants which use technology from bioenergy and hydraulic sources present higher growth, although their contribution is modest.

### 3.2.2. CVF Population Model

Electricity demand determined using CVF population model is presented disaggregated between fossil fuel and clean fuels of energy supply the in analyzed period in Figure 14. Total electricity supply is presented by (T), electricity supply from fossil fuels are represented by (F) and electricity supply from clean fuels are represented by (C). Total electricity supply presents positive a growth behavior meanwhile electricity supply from fossil technologies present an opposite trend and at the same time, clean technologies present a constant growth meaning main energy technologies to be employed are from clean energy sources.

Electricity supply for demand from clean energy sources presents a growth rate of 310.24%, meanwhile electricity demand from fossil energy sources presents a decrease of 5.70% at the end of analyzed period respect year 2010. Same aspect for F values close to 0 were explained on Figure 7.

Electricity demand of 2.0 KWh per capita, shows electricity to supply in analyzed period starts with 2.60 TWh until it raises 3.98 TWh impacting directly in the number of electric power plants generation necessary to satisfy this demand, presenting a growth rate from 25.21% and 149.78% from fossil and clean energy sources, respectively, in the end of analyzed period.

The case of electricity demand of 4.0 KWh per capita, shows electricity to supply from year 2010 to year 2050 initializing with 5.18 TWh until it raises 7.95 TWh. This energy increment implies a growth rate of optimized power plants of 21.40% and 148.36%, from fossil and clean energy sources, respectively.

Optimized fossil and clean power plants number of electricity demand per capita of 1.9 MWh are presented in Figures 15 and 16. For this condition, optimized total power plants are calculated to be 313, from which 99 are planned to be from fossil technology and the rest from clean technology.

The optimized power plants' number from fossil fuel sources behaves quite similar to results obtained when the LVF population model is used, that means, power plant number present an almost

constant behavior during all analyzed period, only gas power plants slightly increase. Clean power plants number presents a discrete but constant growth being bioenergy power plants the technology which presents higher growth, although their contribution is modest. Hydraulic is also the technology which presents the greatest contribution in the energy matrix.

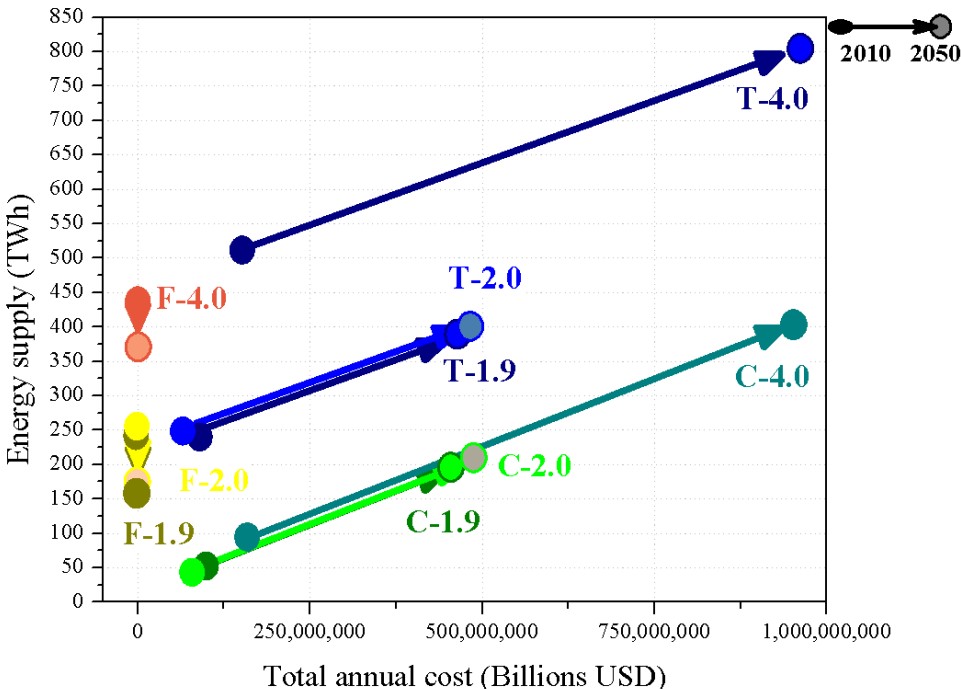

**Figure 14.** Schematic optimized power plants generation centrals number minimizing total annual cost (TAC) of electricity generation for the CVF population model [19] and three electricity demand conditions per capita. Prepared by the authors based on data from own results.

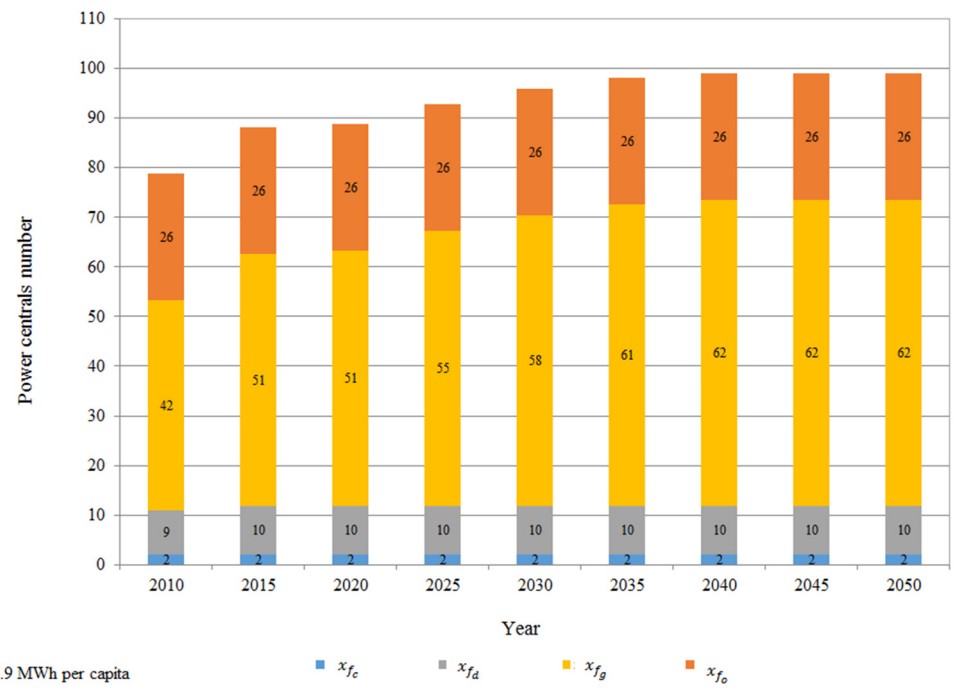

**Figure 15.** Optimized power plants number from fossil energy resource for 1.9 MWh per capita demand with the CVF model.

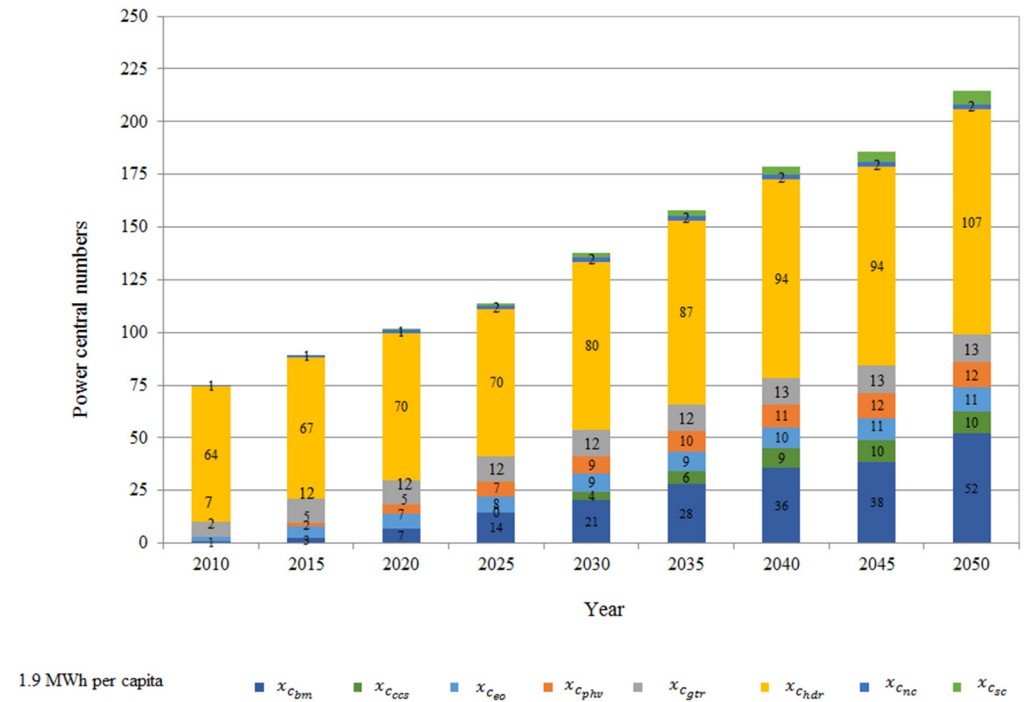

**Figure 16.** Optimized power plants number from clean energy resource for 1.9 MWh per capita demand with the CVF model.

To satisfy electricity demand of 2.0 MWh per capita, required optimized fossil and clean power plants number are presented in Figures 17 and 18.

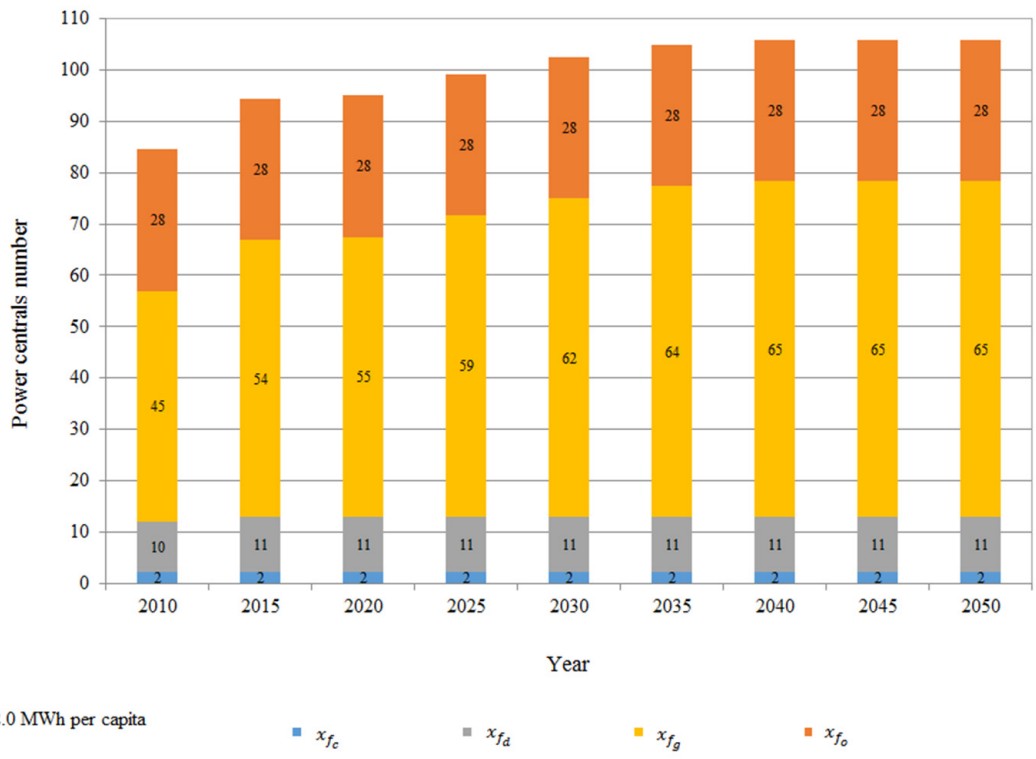

**Figure 17.** Optimized power plants number from fossil energy resource for 2.0 MWh per capita demand with the CVF model.

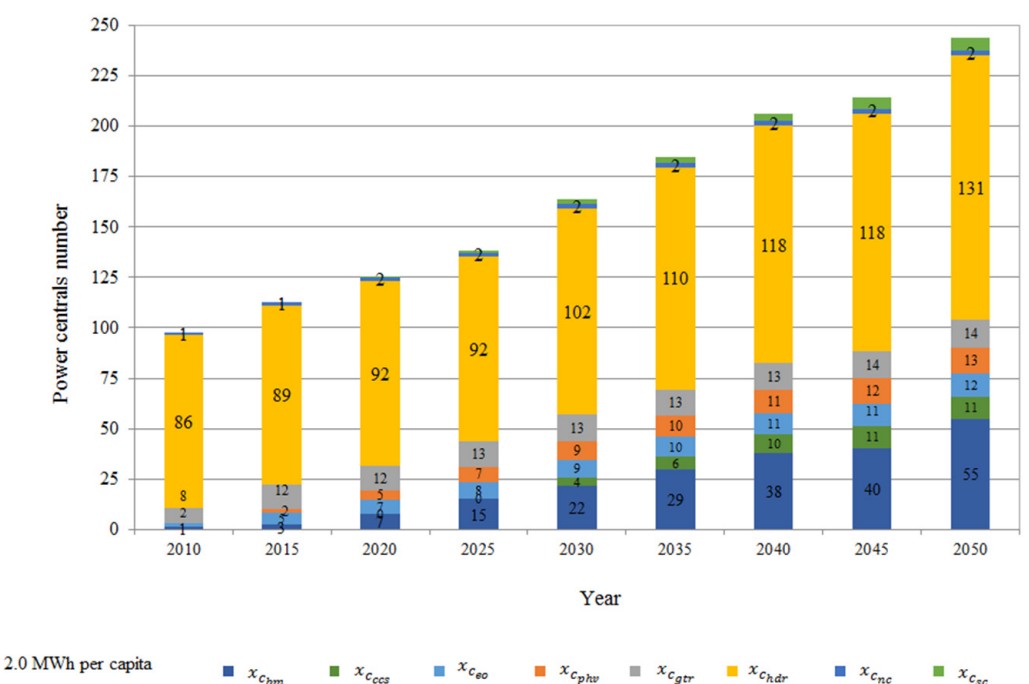

**Figure 18.** Optimized power plants number from clean energy resource for 2.0 MWh per capita demand with the CVF model.

Electricity demand of 4.0 MWh per capita, required optimized fossil and clean power plants number are presented in Figures 19 and 20.

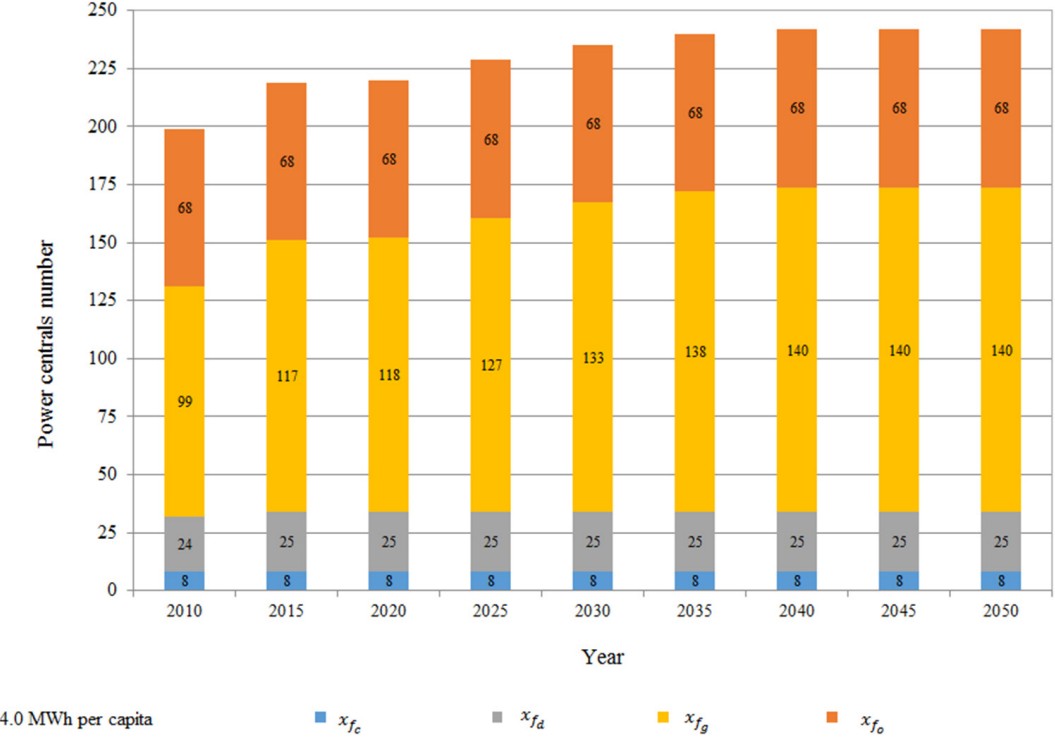

**Figure 19.** Optimized power plants number from fossil energy resource for 4.0 MWh per capita demand with the CVF model.

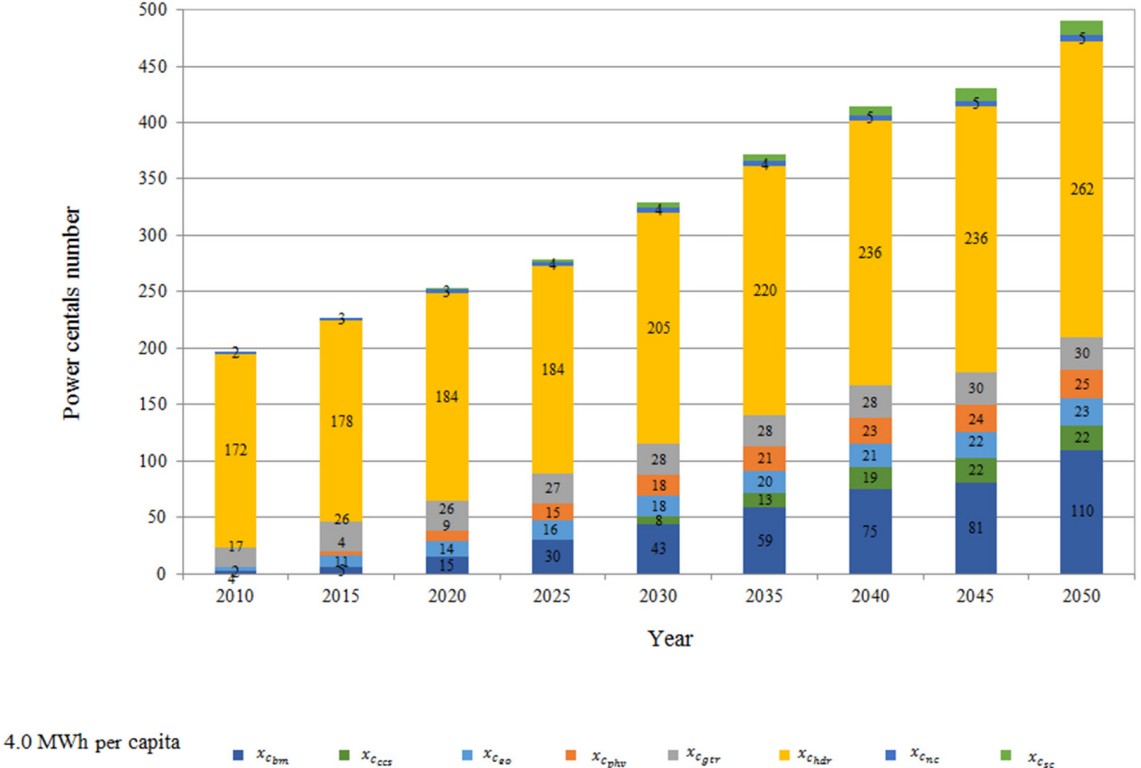

**Figure 20.** Optimized power plants number from clean energy resource for 4.0 MWh per capita demand with the CVF model.

From Figure 19 a behavior similar to the LVF model is observed which means quasi-constant growth during the analysis period for all technologies except gas. The number of plants that use gas presents an increase almost 50% at the end of the period compared to the initial number. The complement of the matrix of optimized power plants from clean plants is presented in Figure 20. As it is observed, clean power number presents constant growth during analyzed period. Power plants which use bioenergy technology present highest growth, with a modest contribution. The energy source with the greatest contribution within the energy matrix corresponds to the hydraulic resource.

It is important to bear in mind that the determined power plant number corresponds to the required plants in each analyzed period, taking as references the number of plants installed at the beginning of the analysis.

Each optimized TAC of electricity power generation condition, implies the energy matrix definition to satisfy the electricity demand as well as $CO_2$eq emissions intrinsic to this process. $CO_2$eq generated due this process are presented below and complement the proposed scenarios in each analyzed period.

### 3.2.3. $CO_2$eq Emissions

$CO_2$eq emissions generated by optimized number of fossil and clean power plants are presented by energy type energy source—fossil and clean.

Produced emissions using fossil sources as primary energy in electrical energy generation for three population demands are presented in Figure 21.

As shown, $CO_2$eq emissions present a decrease as the analysis period increases, expected behavior given that contribution from this energy type source decreases, satisfying objectives proposed in the General Climate Change Law (GCCL) [10].

The corresponding emissions produced by clean sources as primary energy in electricity generation are presented in Figure 22.

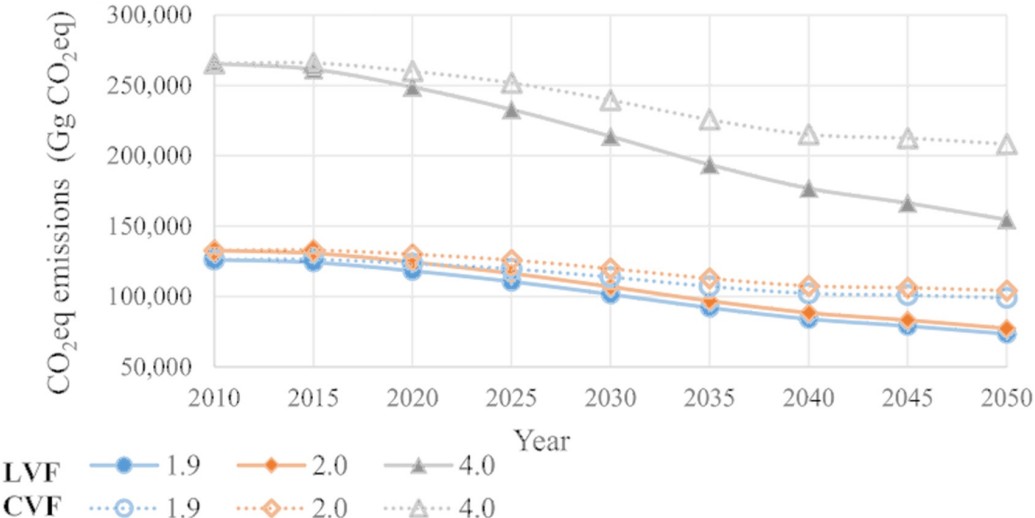

**Figure 21.** $CO_2$eq emissions generated by fossil central power plants for three electricity demand per capita and two population models. Prepared by the authors based on own results.

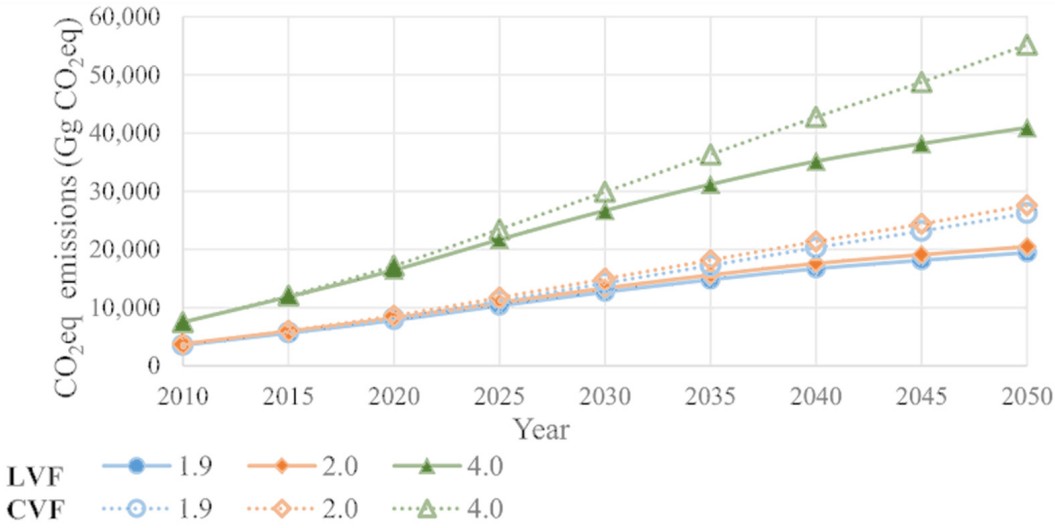

**Figure 22.** $CO_2$eq emissions generated by clean central power plants for three electricity demand per capita and two population models. Prepared by the authors based on own results.

$CO_2$eq emissions generated with clean technologies show an inverse behavior of emissions generated from fossil fuels, that means increasing directly with the period of analysis, however, it is very important to note the amount of emissions generated with this type of energy is, at least, three magnitude orders lower than emissions from fossil sources, satisfying the objective proposed in the General Climate Change Law, GCCL [11].

## 4. Discussion

Energy planning to explore possible alternative scenarios is a necessary tool for the economic development of the country in the face of global, regional events and international commitments acquired in the face of climate change. This tool serves to guide those responsible for energy policy and regulators in the development of policies to visualize scenarios of energy development that allow to contribute effectively to the sustainable growth of the region as well as to demonstrate the impact of policies and plans established in the short and long term. For two grow population model the Figures 23 and 24 are plotted as follow. The data is obtained from the detailed model in Section 2.2.

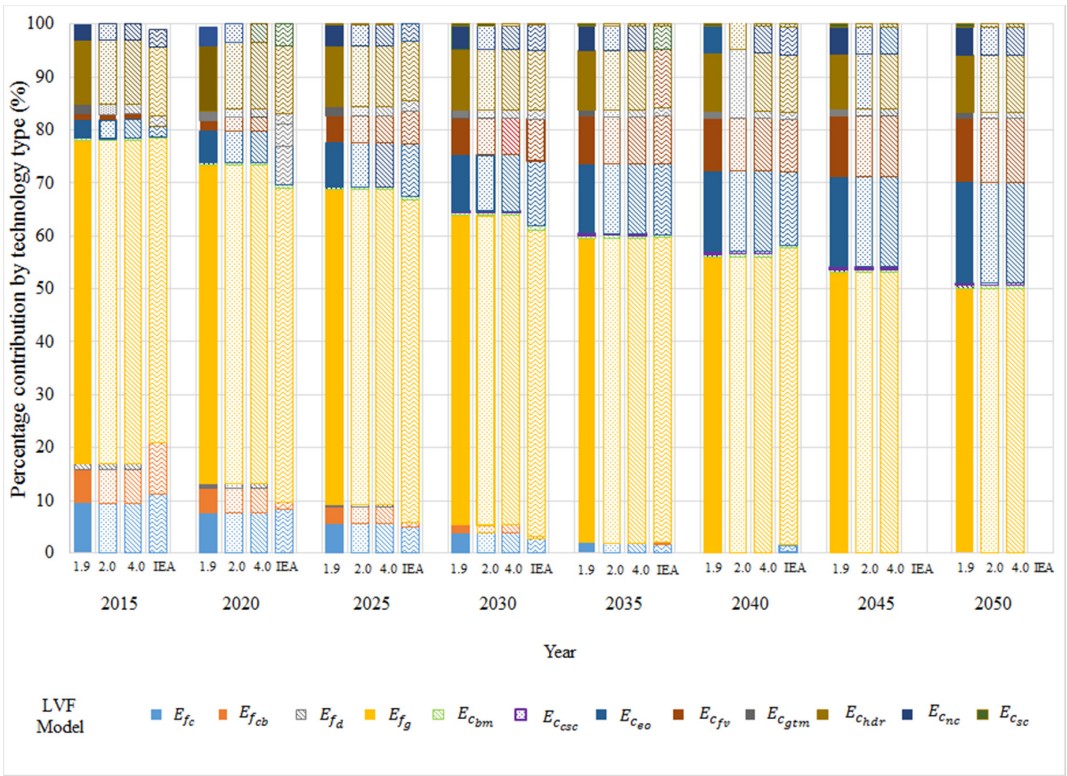

**Figure 23.** LVF Energy superstructure scenario satisfying governmental objectives from GCCL and National Climate Change Strategy (NCCS) [11,12] compared with IEA scenarios. Prepared by the authors based on own results and public data from cited Reference [18].

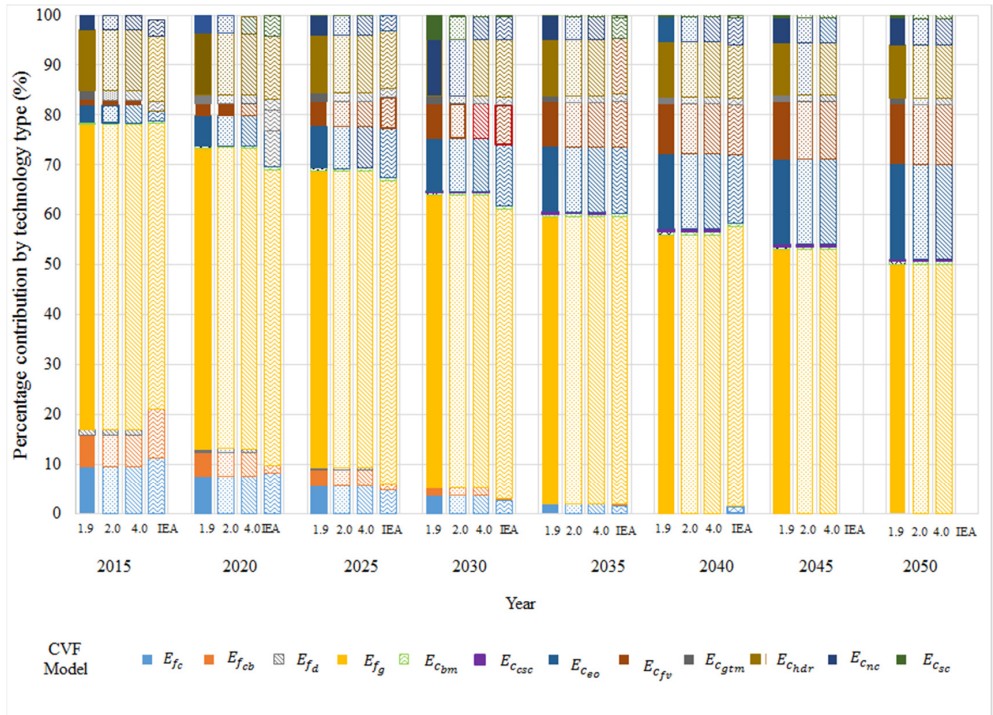

**Figure 24.** CVF Energy superstructure scenario satisfying governmental objectives from GCCL and NCCS [11,12] compared with IEA scenarios. Prepared by the authors based on own results and public data from cited Reference [18].

In these Figures 23 and 24, for the 2015 year, all scenarios take 10% approximately for carbon transformation in the bottom of the columns. The oil plants are close 5% for 1.9, 2.0 and 4.0% except IEA. For IEA in 2015 about 10% was reported. Diesel plants are very low, close to 1% and for IEA it is not reported. The biggest contribution for this scenarios, including IEA is gas process plant, this model has a 61.2% value and IEA is 57.3%. For clean source the values are 21.9 calculated for this model and 20.7% for IEA data.

The maximum deviations for the gas process plant are from 6.3 to 6.8% based on this modelling and IEA reports in 2015. For that year the clean energy variation is from 5.4 to 5.7% variation.

On 2020 calculation, the fossil energy is 73.5%, 73.5% and 73.8% for 1.9, 2.0 and 4.0 scenarios respectively. For IEA projection, fossil energy is 69.0%. The maximum deviation goes from 6.1 to 6.5%. For clean energy, the values are 26.5, 26.5 and 26.2 for 1.9, 2.0 and 4.0 scenarios respectively. But IEA the predicted value is 31.0%. This mean the largest deviation value for our modelling: 14.5% to 16.9% variation.

On 2025, the fossil energy is computed as 68.7% for any scenario and for IEA is 66.8%. This is a variation close to 2.75%. Clean energy values are 31.3% from this work and 33.2% from IEA. This is a variation from 5.7 to 6.0%.

On 2030 predictions, the values for fossil energy the value is constant at 63.8% and for IEA 61.1%. This variation is just from 4.2 to 4.4% for all scenarios. Clean energy for that year is 36.2% compared with IEA 38.9%. Those values have been 6.9% to 7.4% lower.

For 2035 compared values, fossil energy is 59.5% from our model, compared with 59.6% for IEA prediction. They are almost the same value (0.1% variation). This is similar behavior for clean energy—our modelling reports 40.5 and IEA 40.4 (0.2% variation).

Predictions for 2040 for fossil energy must be 55.9% compared with 57.6% from IEA. These values represent 2.95% variation. Clean energy are 44.1% and 42.4% for this work and IEA, respectively. The variation is close to 3.9%.

The IEA do not report for 2045 and 2050. The values for fossil energy are 52.9% and 49.9% for these years. Then the clean energy 47.1 and 50.1% for same years, respectively. This is the goal for NCCS.

There are considerations in each model described in Section 3.2 and a consequence of these, the resulting data are not the same.

Like other optimized models, there are disadvantages, for example dependency of macroeconomic circumstances such as technology cost trends, as well as demographic dependency population but this dependency is inherent to the modelling process.

## 5. Conclusions

This paper has proposed a model that satisfactorily defines energy scenarios integrated by the optimized matrix of power plants required to satisfy the electricity demand, by optimized costs of electricity generation as well as $CO_2$eq emissions produced by this process. In each analysis scenario, governmental objectives with impact on energy security, economy and environmental sustainability are reached satisfactory.

Electricity demand presents a constant growth with both models and three conditions for almost all the period of analysis. This growth is reflected in the gradual increase in the number of plants needed to meet demand. Power plants which use fossil technologies, specifically gas, are those that increase, the rest kept constant. Power plants that use clean source technologies, show a gradual growth, highlighting hydraulics and biomass. The results indicate, in each scenario analyzed, $CO_2$eq emissions satisfy objectives proposed in the General Climate Change Law [11], thus, despite the increase in electricity demand, $CO_2$eq emissions of a reverse behavior.

With the application of the proposed model to the MES, scenarios obtained from applying restricted public policies are visualized. In this scenarios each optimal conditions to achieve goals and plans established in the short and long term are visualized. Given the results obtained with this proposed model, bases are laid with possibility of using the model to identify alternative scenarios to

the optimal ones as well as to evaluate costs of environmental impacts of the technologies in different stages of modelling.

This modelling would be implemented for another country's conditions with few data to predict good agreement with IEA long-term predictions. For the Mexico case, the energy superstructures percentage contribution for 2025 and forward years shows a 7.4% variation or lower compared with this work and IEA data.

**Author Contributions:** G.H.-L., J.M.P.-O. and G.D.T.V. were working in GAMS programming for this Project. R.J.R., A.R.-M., J.C.R. analyzed the data from the Mexican electricity system.

**Funding:** The first author is grateful for the support from the SENER-CONACyT/Energy Sustainability fund for the grant awarded for the postdoctoral stay at the Engineering and Applied Sciences Center Research (CIICAp) of the Autonomous University of Morelos State (UAEM) as well as the support from the Thematic Network on Energy Sustainability, Environment and Society (SUMAS Network), CONACyT project 281101.

**Conflicts of Interest:** The authors declare no conflicts of interest. All figures were prepared by the authors based on own results and public data from cited References.

## Appendix A

The GAMS © code is shown as follow.

| VARIABLES EFdemand, TAC; POSITIVE VARIABLES | |
|---|---|
| Energy contributions (MWh) | |
| Ffos | FOSSIL |
| Frenv | CLEAN |
| Ffoscarb | CARBON |
| Ffoscomb | FUEL OIL |
| Ffosdis | DIESEL |
| Ffosgas | GAS |
| Frenvbio | BIOMASS |
| Frenvcscrb | CARBON CAPTURE AND STORAGE |
| Frenveol | EOLIC |
| Frenvftv | PHOTOVOLTAIC |
| Frenvgtm | GEOTHERMIC |
| Frenvhdr | HYDRAULIC |
| Frenvnucl | NUCLEAR |
| Frenvsc | SOLAR CONCENTRATION |
| *********************************COSTS********************************************* | |
| TAC | TOTAL ANNUAL COST FROM FOSSIL AND CLEAN |
| Costfos | FOSSIL |
| Costrenv | CLEAN |
| Costfoscarb | CARBON |
| Costfoscomb | FUEL OIL |
| Costfosdis | DIESEL |
| Costfosgas | GAS |

| Costrenvbio | BIOMASS |
|---|---|
| Costrenvcscrb | CARBON CAPTURE AND STORAGE |
| Costrenveol | EOLIC |
| Costrenvftv | PHOTOVOLTAIC |
| Costrenvgtm | GEOTHERMIC |
| Costrenvhdr | HYDRAULIC |
| Costrenvnucl | NUCLEAR |
| Costrenvftrm | SOLAR CONCENTRATION |
| ***************************CO$_2$ Emission (kg/MWh)********************************************* | |
| EFdeman | TOTAL EMISSION FROM FOSSIL AND CLEAN |
| EFfos | FOSSIL |
| EFrenv | CLEAN |
| EFfoscarb | CARBON |
| EFfoscomb | OIL FUEL |
| EFfosdis | DIESEL |
| EFfosgas | GAS |
| EFrenvbio | BIOMASS |
| EFrenvcscrb | CARBON CAPTURE AND STORAGE |
| EFrenveol | EOLIC |
| EFrenvftv | PHOTOVOLTAIC |
| EFrenvgtm | GEOTHERMIC |
| EFrenvhdr | HYDRAULIC |
| EFrenvnucl | NUCLEAR |
| EFrenvftrm | SOLAR CONCENTRATION |
| ******************* NUMBER OF POWER PLANTS GENERATION ********************************* | |
| xtot | TOTAL |
| xf | FOSSILS |
| xr | CLEAN |
| xfoscarb | CARBON |
| xfoscomb | FUEL OIL |
| xfosdis | DIESEL |
| xfosgas | GAS |
| xrenvbio | BIOMASA |
| xrenvcscrb | CARBON CAPTURE AND STORAGE |
| xrenveol | EOLICA |
| xrenvftv | PHOTOVOLTAIC |
| xrenvgtm | GEOTHERMIC |
| xrenvhdr | HYDRAULIC |
| xrenvnucl | NUCLEAR |
| xrenvftrm | SOLAR CONCENTRATION |

| PARAMETERS | | |
|---|---|---|
| ***********************************COSTS (USD/MWh neto)*********************************** | | |
| ************************* INVESTMENT COSTS BY ENERGY FUEL************************************ | | |
| afoscarb | CARBON | /37.39/ |
| afoscomb | OIL FUEL | /41.891/ |
| afosdis | DIESEL | /64.16/ |
| afosgas | GAS | /72.53/ |
| arenvbio | BIOMASS | /150/ |
| arenvcscrb | CARBON CAPTURE AND STORAGE | /47.39/ |
| arenveol | EOLIC | /1.65E6/ |
| arenvftv | PHOTOVOLTAIC | /5.11E6/ |
| arenvgtm | GEOTHERMIC | /3.18E6/ |
| arenvhdr | HYDRAULIC | /2.0E6/ |
| arenvnucl | NUCLEAR | /87.93/ |
| arenvftrm | SOLAR CONCENTRATION | /180/ |
| *************************** ASSOCIATED COST FROM FUEL TYPE ******************************** | | |
| bfoscarb | CARBON | /27.57/ |
| bfoscomb | OIL FUEL | /93.89/ |
| bfosdis | DIESEL | /147.06/ |
| bfosgas | GAS | /49.37/ |
| brenvbio | BIOMASS | /7.9/ |
| brenvcscrb | CARBON CAPTURE AND STORAGE | /27.67/ |
| brenveol | EOLICA | /0/ |
| brenvftv | PHOTOVOLTAIC | /0/ |
| brenvgtm | GEOTHERMIC | /50.12/ |
| brenvhdr | HYDRAULIC | /6.38/ |
| brenvftrm | SOLAR CONCENTRATION | /0/ |
| OPERATION AND MAINTENANCE COST, INCLUDING FIXED AND VARIABLE COSTS * | | |
| cfoscarb | CARBON | /8.13/ |
| cfoscomb | OIL FUEL | /9.21/ |
| cfosdis | DIESEL | /12.88/ |
| cfosgas | GAS | /11.33/ |
| crenvbio | BIOMAS | /4.33/ |
| crenvcscrb | CARBON CAPTURE AND STORAGE | /8.13/ |
| crenveol | EOLIC | /1.75/ |
| crenvftv | PHOTOVOLTAIC | /7.72/ |
| crenvgtm | GEOTHERMIC | /19.41/ |
| crenvhdr | HYDRAULICA | /5.71/ |
| crenvftrm | SOLAR CONCENTRATION | /20/ |

| ********************* CO₂eq BY FUEL (Kg CO₂/MWh)********************* | | |
|---|---|---|
| ECO2eqfoscarb | CARBON | /1089.81/ |
| ECO2eqfoscomb | OIL FUEL | /822/ |
| ECO2eqfosdis | DIESEL | /274.44/ |
| ECO2eqfosgas | GAS | /524/ |
| ECO2eqrenvbio | BIOMAS | /1403.75/ |
| ECO2eqrenvcscrb | CARBON CAPTURE AND STORAGE | /217.2/ |
| ECO2eqrenveol | EOLIC | /210/ |
| ECO2eqrenvftv | PHOTOVOLTAIC | /106/ |
| ECO2eqrenvgtm | GEOTHERMIC | /372/ |
| ECO2eqrenvhdr | HYDRAULIC | /15/ |
| ECO2eqrenvftrm | SOLAR CONCENTRATION | /14/ |
| Fdemand | ANNUAL NATIONAL ELECTRICITY DEMAND MWh | /246381740/ |

| * ELECTRICITY CONTRIBUTION GENERATED IN A PERIOD PRIOR TO THE ANALYZED (MWh)* | | |
|---|---|---|
| Ffcarbins | CARBON | /18380000/ |
| Ffcombins | OIL FUEL | /6451200/ |
| Ffdisins | DIESEL | /780000/ |
| Ffgasins | GAS | /27166000/ |
| Frbioins | BIOMAS | /0/ |
| Frcscrbins | CARBON CAPTURE AND STORAGE | /0/ |
| Freolins | EOLIC | /5000/ |
| Frftvins | PHOTOVOLTAIC | /00/ |
| Frgtmins | GEOTHERMIC | /729900/ |
| Frhdrins | HYDRAULIC | /26851000/ |
| Frnuclins | NUCLEAR | /1080500/ |
| Frftrmins | SOLAR CONCENTRATION | /0/ |

| ************************* CAPACITY OF POWER PLANT (MW)************************************* | | |
|---|---|---|
| Fcapfoscarb | CARBON | /2600/ |
| Fcapfoscomb | OIL FUEL | /12711/ |
| Fcapfosdis | DIESEL | /182/ |
| Fcapfosgas | GAS | /7230/ |
| Fcaprenvbio | BIOMASA | /1500/ |
| Fcaprenvcscrb | CARBON CAPTURE AND STORAGE | /1000/ |
| Fcaprenveol | EOLIC | /20000/ |
| Fcaprenvftv | PHOTOVOLTAIC | /1000/ |
| Fcaprenvgtm | GEOTHERMIC | /1000/ |
| Fcaprenvhdr | HYDRAULIC | /10270/ |
| Fcaprenvnucl | NUCLEAR | /1365/ |
| Fcaprenvftrm | SOLAR CONCENTRATION | /960/ |

| EQUATIONS | |
|---|---|
| R1 | ANNUAL ELECTRICY DEMAND |
| R2 | FOSSIL FUEL CONTRUIBUTION TO ELECTRICITY DEMAND |
| R3 | CLEAN FUEL CONTRUIBUTION TO ELECTRICITY DEMAND |
| R4 | ELECTRICITY GENERATION FROM CARBON |
| R5 | ELECTRICITY GENERATION FROM FUEL OIL |
| R6 | ELECTRICITY GENERATION FROM DIESEL |
| R7 | ELECTRICITY GENERATION FROM GAS |
| R8 | ELECTRICITY GENERATION FROM BIOMAS |
| R9 | ELECTRICITY GENERATION FROM CARBON CAPTURE AND STORAGE |
| R10 | ELECTRICITY GENERATION FROM EOLIC |
| R11 | ELECTRICITY GENERATION FROM PHOTOVOLTAIC |
| R12 | ELECTRICITY GENERATION FROM GEOTHERMAL |
| R13 | ELECTRICITY GENERATION FROM HYDRAULIC |
| R14 | ELECTRICITY GENERATION FROM NUCLEAR |
| R15 | ELECTRICITY GENERATION FROM SOLAR CONCENTRATION |
| ***********************************COSTS*********************************** | |
| R16 | TOTAL ANNUAL COST OF ELECTRICITY GENERATION |
| R17 | TOTAL ANNUAL COST OF ELECTRICITY GENERATION FROM FOSSIL FUELS |
| R18 | TOTAL ANNUAL COST OF ELECTRICITY GENERATION FROM CLEAN FUELS |
| R19 | TOTAL ANNUAL COST OF ELECTRICITY GENERATION FROM CARBON |
| R20 | TOTAL ANNUAL COST OF ELECTRICITY GENERATION FROM OIL FUEL |
| R21 | TOTAL ANNUAL COST OF ELECTRICITY GENERATION FROM DIESEL |
| R22 | TOTAL ANNUAL COST OF ELECTRICITY GENERATION FROM GAS |
| R23 | TOTAL ANNUAL COST OF ELECTRICITY GENERATION FROM BIOMAS |
| R24 | TOTAL ANNUAL COST OF ELECTRICITY GENERATION FROM CARBON CAPTURE AND STORAGE |
| R25 | TOTAL ANNUAL COST OF ELECTRICITY GENERATION FROM EOLIC |
| R26 | TOTAL ANNUAL COST OF ELECTRICITY GENERATION FROM PHOTOVOLTAIC |
| R27 | TOTAL ANNUAL COST OF ELECTRICITY GENERATION FROM GEOTHERMAL |
| R28 | TOTAL ANNUAL COST OF ELECTRICITY GENERATION FROM HYDRAULIC |
| R29 | TOTAL ANNUAL COST OF ELECTRICITY GENERATION FROM NUCLEAR |
| R30 | TOTAL ANNUAL COST OF ELECTRICITY GENERATION FROM SOLAR CONCENTRATION |
| **********************************$CO_2$eq EMISSIONS********************************** | |
| R31 | CO2eq EMISSION FROM TOTAL ELECTRICITY GENERATION |
| R32 | CO2eq EMISSION FROM ELECTRICITY GENERATION FROM FOSSIL FUELS |
| R33 | CO2eq EMISSION FROM ELECTRICITY GENERATION FROM CLEAN FUELS |
| R34 | CO2eq EMISSION FROM ELECTRICITY GENERATION FROM CARBON |
| R35 | CO2eq EMISSION FROM ELECTRICITY GENERATION FROM FUEL OIL |
| R36 | CO2eq EMISSION FROM ELECTRICITY GENERATION FROM DIESEL |

| R37 | CO2eq EMISSION FROM ELECTRICITY GENERATION FROM GAS |
|-----|------|
| R38 | CO2eq EMISSION FROM ELECTRICITY GENERATION FROM BIOMAS |
| R39 | CO2eq EMISSION FROM ELECTRICITY GENERATION FROM CARBON CAPTURE AND STORAGE |
| R40 | CO2eq EMISSION FROM ELECTRICITY GENERATION EOLIC |
| R41 | CO2eq EMISSION FROM ELECTRICITY GENERATION PHOTOVOLTAIC |
| R42 | CO2eq EMISSION FROM ELECTRICITY GENERATION GEOTHERMIC |
| R43 | CO2eq EMISSION FROM ELECTRICITY GENERATION HYDRAULIC |
| R44 | CO2eq EMISSION FROM ELECTRICITY GENERATION NUCLEAR |
| R45 | CO2eq EMISSION FROM ELECTRICITY GENERATION SOLAR CONCENTRATION |
| **********************************RESTRICTIONS********************************** | |
| R46 | ELECTRICITY CONTRIBUTION FROM CARBON |
| R47 | ELECTRICITY CONTRIBUTION FROM FUEL OIL |
| R48 | ELECTRICITY CONTRIBUTION FROM DIESEL |
| R49 | ELECTRICITY CONTRIBUTION FROM GAS |
| R50 | ELECTRICITY CONTRIBUTION FROM BIOMAS |
| R51 | ELECTRICITY CONTRIBUTION FROM CARBON CAPTURE AND STORAGE |
| R52 | ELECTRICITY CONTRIBUTION FROM EOLIC |
| R53 | ELECTRICITY CONTRIBUTION FROM PHOTOVOLTAIC |
| R54 | ELECTRICITY CONTRIBUTION FROM GEOTHERMIC |
| R55 | ELECTRICITY CONTRIBUTION FROM HYDRAULIC |
| R56 | ELECTRICITY CONTRIBUTION FROM NUCLEAR |
| R57 | ELECTRICITY CONTRIBUTION FROM SOLAR CONCENTRATION |
| R58 | ELECTRICITY CONTRIBUTION FROM FOSSIL FUELS |
| R59 | TOTAL NUMBER OF POWER PLANTS GENERATION |
| R60 | FOSSIL NUMBER OF POWER PLANTS GENERATION |
| R61 | CLEAN NUMBER OF POWER PLANTS GENERATION |

| *******************************************MODEL*************************************************** | |
|------|------|
| *******************************************ELECTRICIY CALCULUS *********************************************** | |
| R1.. | Fdemand = E = Ffos + Frenv; |
| R2.. | Ffos = E = Ffoscarb + Ffoscomb + Ffosdis + Ffosgas; |
| R3.. | Frenv = E = Frenvbio + Frenvcscrb + Frenveol + Frenvftv + Frenvftrm + Frenvgtm + Frenvhdr + Frenvnucl; |
| *************************FÓSILES******************************************************************** | |
| R4.. | Ffoscarb =L= Ffcarbins + ( Fcapfoscarb * xfoscarb * 8760 * fafoscarb); |
| R5.. | Ffoscomb =L= Ffcombins + ( Fcapfoscomb * xfoscomb * 8760 * fafoscomb); |
| R6.. | Ffosdis =L= Ffdisins + ( Fcapfosdis * xfosdis * 8760 * fafosdis); |
| R7.. | Ffosgas =L= Ffgasins + ( Fcapfosgas * xfosgas * 8760 * fafosgas); |

| | |
|---|---|
| \***************************\*RENOVABLES\******************************************************************* | |
| R8.. | Frenvbio =L= Frbioins + ( Fcaprenvbio * xrenvbio * 8760 * farenvbio) ; |
| R9.. | Frenvcscrb =L= Frcscrbins + ( Fcaprenvcscrb * xrenvcscrb * 8760 * farenvcscrb) ; |
| R10.. | Frenveol =L= Freolins + ( Fcaprenveol * xrenveol * 8760 * farenveol) ; |
| R11.. | Frenvftv =L= Frftvins + ( Fcaprenvftv * xrenvftv * 8760 * farenvftv) ; |
| R12.. | Frenvftrm =L= Frftrmins + ( Fcaprenvftrm * xrenvftrm * 8760 * farenvftrm) ; |
| R13.. | Frenvgtm =L= Frgtmins + ( Fcaprenvgtm * xrenvgtm * 8760 * farenvgtm) ; |
| R14.. | Frenvhdr =L= Frhdrins + ( Fcaprenvhdr * xrenvhdr * 8760 * farenvhdr) ; |
| R15.. | Frenvnucl =L= Frnuclins + ( Fcaprenvnucl * xrenvnucl * 8760 * farenvnucl) ; |
| \******************************************************************************* | |
| \***************************\*COST CALCULUS\*************************************************************** | |
| R16.. | TAC =E= Costfos + Costrenv ; |
| R17.. | Costfos =E= Costfoscarb + Costfoscomb + Costfosdis + Costfosgas ; |
| R18.. | Costrenv =E= Costrenvbio + Costrenvcscrb + Costrenveol + Costrenvftv + Costrenvftrm + Costrenvgtm + Costrenvhdr + Costrenvnucl; |
| R19.. | Costfoscarb =E= Ffoscarb * (afoscarb + bfoscarb + cfoscarb) * 1000 ; |
| R20.. | Costfoscomb =E= Ffoscomb * (afoscomb + bfoscomb + cfoscomb) * 1000 ; |
| R21.. | Costfosdis =E= Ffosdis * (afosdis + bfosdis + cfosdis) * 1000 ; |
| R22.. | Costfosgas =E= Ffosgas * (afosgas + bfosgas + cfosgas) * 1000 ; |
| R23.. | Costrenvbio =E= Frenvbio * (arenvbio + brenvbio + crenvbio) * 1000 ; |
| R24.. | Costrenvcscrb =E= Frenvcscrb * (arenvcscrb + brenvcscrb + crenvcscrb) * 1000 ; |
| R25.. | Costrenveol =E= Frenveol * (arenveol + brenveol + crenveol) * 1000 ; |
| R26.. | Costrenvftv =E= Frenvftv * (arenvftv + brenvftv + crenvftv) * 1000 ; |
| R27.. | Costrenvgtm =E= Frenvgtm * (arenvgtm + brenvgtm + crenvgtm) * 1000 ; |
| R28.. | Costrenvhdr =E= Frenvhdr * (arenvhdr + brenvhdr + crenvhdr) * 1000 ; |
| R29.. | Costrenvnucl =E= Frenvnucl * (arenvnucl + brenvnucl + crenvnucl) * 1000 |
| R30.. | Costrenvftrm =E= Frenvftrm * (arenvftrm + brenvftrm + crenvftrm) * 1000 ; |
| \******************************************************************************* | |
| R31.. | EFdemand =E= EFfos + EFrenv; |
| \******************************************************************************* | |
| \*************************** EMISSIONS CALCULUS\*********************************************** | |
| R32.. | EFfos =E= EFfoscarb + EFfoscomb + EFfosdis + EFfosgas ; |
| R33.. | EFrenv =E= EFrenvbio + EFrenvcscrb + EFrenveol + EFrenvftv + EFrenvftrm + EFrenvgtm + EFrenvhdr; |
| \***************************\*DE FUENTES FÓSILES\*********************************************************** | |
| R34.. | EFfoscarb =E= ECO2eqfoscarb * Ffoscarb ; |
| R35.. | EFfoscomb =E= ECO2eqfoscomb * Ffoscomb ; |
| R36.. | EFfosdis =E= ECO2eqfosdis * Ffosdis ; |
| R37.. | EFfosgas =E= ECO2eqfosgas * Ffosgas ; |

| | |
|---|---|
| ***************************DE FUENTES RENOVABLES************************************************* | |
| R38.. | EFrenvbio =E= ECO2eqrenvbio * Frenvbio ; |
| R39.. | EFrenvcscrb =E= ECO2eqrenvcscrb * Frenvcscrb ; |
| R40.. | EFrenveol =E= ECO2eqrenveol * Frenveol ; |
| R41.. | EFrenvftv =E= ECO2eqrenvftv * Frenvftv ; |
| R42.. | EFrenvftrm =E= ECO2eqrenvftrm * Frenvftrm ; |
| R43.. | EFrenvgtm =E= ECO2eqrenvgtm * Frenvgtm ; |
| R44.. | EFrenvhdr =E= ECO2eqrenvhdr * Frenvhdr ; |
| R45.. | EFrenvnucl =E= ECO2eqrenvnucl * Frenvnucl ; |
| ************************PERCENTAGE TECHNOLOGY CONTRIBUTIONS************************ | |
| R46.. | Ffoscarb =L= Ffos ; |
| R47.. | Ffoscomb =L= Ffos ; |
| R48.. | Ffosdis =L= Ffos ; |
| R49.. | Ffosgas =L= Ffos ; |
| R50.. | Frenvbio =L= Fdemand - Ffos ; |
| R51.. | Frenvcscrb =L= Fdemand - Ffos ; |
| R52.. | Frenveol =L= Fdemand - Ffos ; |
| R53.. | Frenvftv =L= Fdemand - Ffos ; |
| R54.. | Frenvftrm =L= Fdemand - Ffos ; |
| R55.. | Frenvgtm =L= Fdemand - Ffos ; |
| R56.. | Frenvhdr =L= Fdemand - Ffos ; |
| R57.. | Frenvnucl =L= Fdemand - Ffos ; |
| ***********FOSSIL PERCENTAGE CONTRIBUTION *********** | |
| R58.. | Ffos =L= Fdemand * 0.8132 ; |
| **************************CENTTRAL NUMERO DE CENTRALES*************************** | |
| R59.. | xtot =E= xf + xr ; |
| R60.. | Xf =E= xfoscarb + xfoscomb + xfosdis + xfosgas ; |
| R61.. | xr =E= xrenvbio + xrenvcscrb + xrenveol + xrenvftv + xrenvftrm + xrenvgtm + xrenvhdr + xrenvnucl; |
| *********************NUMBER OF POWER PLANTS GENERATION ***************************************** | |
| | Ffoscarb.LO = Fdemand * 0.11 ; |
| | Ffoscarb.UP = Fdemand * 0.115 ; |
| | Ffoscomb.LO = Fdemand * 0.075 ; |
| | Ffoscomb.UP = Fdemand * 0.0781 ; |
| | Ffosdis.LO = Fdemand * 0.008 ; |
| | Ffosdis.UP = Fdemand * 0.01 ; |
| | Ffosgas.LO = Fdemand * 0.45 ; |
| | Ffosgas.UP = Fdemand * 0.61 ; |
| | Frenvbio.LO = Fdemand * 0.00 ; |

| | |
|---|---|
| | Frenvbio.UP = Fdemand * 0.003 ; |
| | Frenvcscrb.LO = Fdemand * 0.00 ; |
| | Frenvcscrb.UP = Fdemand * 0.00 ; |
| | Frenveol.LO = Fdemand * 0.190 ; |
| | Frenveol.UP = Fdemand * 0.013 ; |
| | Frenvftrm.LO = Fdemand * 0.00 ; |
| | Frenvftrm.UP = Fdemand * 0.00 ; |
| | Frenvftv.LO = Fdemand * 0.0055 ; |
| | Frenvftv.UP = Fdemand * 0.0115 ; |
| | Frenvgtm.LO = Fdemand * 0.0145 ; |
| | Frenvgtm.UP = Fdemand * 0.015 ; |
| | Frenvhdr.LO = Fdemand * 0.11 ; |
| | Frenvhdr.UP = Fdemand * 0.1194 ; |
| | Frenvnucl.LO = Fdemand * 0.05 ; |
| | Frenvnucl.UP = Fdemand * 0.027 ; |
| MODEL PCL05 /ALL/;<br>SOLVE PCL05 USING MIP MINIMIZING TAC; | |

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
