# Peer review of "Energy Model for Long-Term Scenarios in Power Sector under Energy Transition Laws"

_processes, doi:10.3390/pr7100674_

Round 1

Reviewer 1 Report

Prediction of energy demand is an important topic. This paper presents the study on developing an energy model to optimize the grid of power plants of Mexican electricity sector. However, the method presented in this study is very simple. The linear model was developed based on several key indicators from literature without validation. The algorithm of the model was not presented clearly, such as objective function and constraints. The uncertainties of the results cannot be evaluated. Therefore, the scientific contribution of this study is low. 

Other comments are listed as follows.

Existing keywords are very general and could be improved to reflect the key work in this study. In Table 1., why are electricity demands same for 2010, 2015, 2020 for each demand condition ? In Figure 8., why are the total annual costs for all F scenarios 0? There is only 1 short paragraph in Section 4. Discussion. It may not be necessary to set a separated section for this paragraph.

Author Response

Reviewer 1

Q: Prediction of energy demand is an important topic. 

Thank you for your comment.

Q: This paper presents the study on developing an energy model to optimize the grid of power plants of Mexican electricity sector. However, the method presented in this study is very simple. 

Main contribution of this paper is the equation set is based on actual emission source for Mexican scenario. This scenario can be used with constraints for getting a probable long term scenario 

Q: The linear model was developed based on several key indicators from literature without validation. 

The model is based on country oficial data. The population grow is based on the United Nation data. The prediction energy consumption is calculated on those values. The detailed equations are reported in section 2.1   

Q: The algorithm of the model was not presented clearly, such as objective function and constraints. 

A new figure was added to the section 2 (Methods) with a clear description and the GAMS © code is include in appendix A to show the equation set with the constraints and the objective function. 

Q: The uncertainties of the results cannot be evaluated. 

Into the discussion section, the new figures 22 and 23 show the variation from IEA data and our modelling. This values are commented with the percentage variation. The uncertainties are computed and show a bigger deviation for 2020 year prediction. The lowest variation between the IEA data and this work is 2035 year, lower than 0.1 %

Q: Therefore, the scientific contribution of this study is low. 

The scientific contribution is a simplified modelling based on two grow models and three per capita energy consumption with very good agreement with IEA data prediction (lower than 7.4 % variation). This tool would be adapted to any other country with few data to determine the percentage for clean energy plants to get a CO2eq desired objective. 

Q: Existing keywords are very general and could be improved to reflect the key work in this study. 

The keywords were edited. These new keywords are representative of this work.

Q: In Table 1., why are electricity demands same for 2010, 2015, 2020 for each demand condition ? 

The table were cleared with a previous text: Three demand conditions per capita are assumed in each model: 1.9, 2.0 and 4.0 MWh. The demand of 1.9 MWh per capita is taken from the historical demand of Mexico in the period of 2000 to 2010; the demand of 2.0 MWh per capita considers an increase of 10 % in the population demand and finally, the demand of 4.0 MWh per capita is the typical demand of the population consumption from developed countries.

Q:In Figure 8., why are the total annual costs for all F scenarios 0? 

There is not a 0 (it is 2.1E+08) but for the figure scale these are close to that. This mean there is not investment on fossil fuels from 2010 to 2050. This were explained for that figure (now it is figure 7) and for figure 14, also.

Q: There is only 1 short paragraph in Section 4. Discussion. It may not be necessary to set a separated section for this paragraph.

The discussion section was edited with new information. The scenarios were compared with IEA prediction data for Mexico. The percentages energy show a very good agreement for years 2025 and beyond. The variations between this work data and IEA data are lower than 7.4 %. Conclusions sections include this explanation too.

Reviewer 2 Report

Some spell, grammar and expression mistakes must be corrected, like in line 19. 137. 523 etc. Sοme more information-explanation for model formulation should be added. I think that one who reads the paper could be confused due to the presenation.

1. The model which has been presented in "Model formulation" section, needs to be presented more clearly...with more information about eg.  constraints..If they have been used in practice or they are theoritical only.
2.
Some spell, grammar and expression mistakes must be corrected, like in line 19. 137. 523 etc.
3. I have doubts about the method that has been used. Is its simplicity correct? What if the method has been simplified so much that it may not takes into account improtant parameters? (I think so)

Author Response

Reviewer 2

Q: The model which has been presented in "Model formulation" section, needs to be presented more clearly...with more information about eg. constraints..

The scientific contribution is a simplified modelling based on two grow models and three per capita energy consumption with very good agreement with IEA data prediction (lower than 7.4 % variation). This tool would be adapted to any other country with few data to determine the percentage for clean energy plants to get a CO2eq desired objective. A new figure was added to the section 2 (Methods) with a clear description and the GAMS © code is include in appendix A to show the equation set with the constraints and the objective function.

Q: If they have been used in practice or they are theoretical only.

This is a theoretical scenario to explain how much is the Total Annual Cost for clean energy plants in Mexico case.   

Q: 2.Some spell, grammar and expression mistakes must be corrected, like in line 19. 137. 523 etc.

The document was carefully edited to improve spelling and grammar.

Q: 3. I have doubts about the method that has been used. Is its simplicity correct? 

The discussion section was edited with new information to validate this simplification. The scenarios were compared with IEA prediction data for Mexico. The percentages energy show a very good agreement for years 2025 and beyond. The variations between this work data and IEA data are lower than 7.4 %. We think this work has correct results.

Q: What if the method has been simplified so much that it may not takes into account important parameters? (I think so)

Thank you to observe it. This is the main point: The simplification does not derive bigger variation. For long term predictions the percentages may match with the IEA data. 

Q: Sοme more information-explanation for model formulation should be added. 

The appendix A were added. This section has the entire used data for the calculation. A new figure with explanation was added on section 2: methods. 

Q: I think that one who reads the paper could be confused due to the presentation.

Thank you for your comments, the paper now is clear to readers and the added appendix avoid confusion. 

Round 2

Reviewer 1 Report

Most of the comments have been answered and relevant corrections have been made in the revised version. The quality of the paper was improved.

Author Response

Thank you for your comments . 

Reviewer 2 Report

I think that some more discussion-comments have to be done about the deviation percentages...eg. which is a possible reason for these deviations and especially why in 2020 is there a so large deviation? Is it a modelling fault? Based on which reference are the deviation percentages judged ok? What about the disadvantages of the proposed modelling?

Author Response

Comments and Suggestions for Authors

I think that some more discussion-comments have to be done about the deviation percentages...eg. which is a possible reason for these deviations and especially why in 2020 is there a so large deviation?

It is important to mention that the proposed model is an alternative model from the IEA´s energy model. The main difference at the proposed model is an optimization for the electricity matrix generation; IEA´s model uses a large scale simulation for replicate energy markets with global effects. For these reasons, the results prospective from both models would not necessary to be equal. It is important to notice that deviations between both prospective are not so far from each other.

The year 2020 is a divergence for one reason: there is an abrupt reduction of electricity generation from oil, from the year 2014 to 2020 and beyond in the IEA´s model, see Figure 1. The decrease for each year is -1.37 in that period.

Dy/Dx = - 1.37

In our optimized proposed model, the behavior of electricity generation from oil presents a smooth yearly decrease of -0.34, from 2015 to 2020. This value is lower than IEA´s model. This behavior can be appreciated in Figure 2.

Dy/Dx = -0.34

Is it a modelling fault?

Authors do not consider a fault from models. There are considerations in each model described in 3.2 section and a consequence of these, the resulted data are not the same.

This text is now included in the new version into last part discussion section.

Based on which reference are the deviation percentages judged ok? 

Authors compared the IEA projections data with a simplified model data. Reference is not included because the percentages variations were calculated as variation over IEA data.  We would not be able to mention which deviation percentages are ok or not between models. Both models proposed a prospective, the authors’ paper results are not so far from an international energy agency (IEA) reference. The proposed model in this paper could be a useful tool to provide expectations well-founded which can be also an invitation to impulse clean technologies considering México’s reality under international commitments.

What about the disadvantages of the proposed modelling? 

Like others optimized model, there are disadvantages for example dependency of macroeconomic circumstances such as technology cost trends as well as demographic dependency population, but this dependency is inherent to the modelling process.

This text is now included in the new version into last part discussion section.

Other disadvantage factor could be infrastructure’s useful lifetime or energy resources data; if these values would be available for Mexico, these could be included in a detailed future model and this proposed model is a simplified version.
